# A guanidine-degrading enzyme controls genomic stability of ethylene-producing cyanobacteria

Bo Wang [1,2✉], Yao Xu[3], Xin Wang [4,6], Joshua S. Yuan [4], Carl H. Johnson[3], Jamey D. Young [2,5] & Jianping Yu [1✉]

Recent studies have revealed the prevalence and biological significance of guanidine metabolism in nature. However, the metabolic pathways used by microbes to degrade guanidine or mitigate its toxicity have not been widely studied. Here, via comparative proteomics and subsequent experimental validation, we demonstrate that Sll1077, previously annotated as an agmatinase enzyme in the model cyanobacterium *Synechocystis* sp. PCC 6803, is more likely a guanidinase as it can break down guanidine rather than agmatine into urea and ammonium. The model cyanobacterium *Synechococcus elongatus* PCC 7942 strain engineered to express the bacterial ethylene-forming enzyme (EFE) exhibits unstable ethylene production due to toxicity and genomic instability induced by accumulation of the EFE-byproduct guanidine. Co-expression of EFE and Sll1077 significantly enhances genomic stability and enables the resulting strain to achieve sustained high-level ethylene production. These findings expand our knowledge of natural guanidine degradation pathways and demonstrate their biotechnological application to support ethylene bioproduction.

[1] Biosciences Center, National Renewable Energy Laboratory, Golden, CO, USA. [2] Department of Chemical and Biomolecular Engineering, Vanderbilt University, Nashville, TN, USA. [3] Department of Biological Sciences, Vanderbilt University, Nashville, TN, USA. [4] Synthetic and Systems Biology Innovation Hub, Department of Plant Pathology and Microbiology, Texas A&M University, College Station, TX, USA. [5] Department of Molecular Physiology and Biophysics, Vanderbilt University, Nashville, TN, USA. [6] Present address: Department of Microbiology, Miami University, Oxford, OH, USA.
✉email: bo.wang.vu@gmail.com; Jianping.Yu@nrel.gov

Despite the practical applications of guanidine as a protein denaturant (when applied at high concentrations)[1] and as an ingredient in slow-release fertilizers[2], little is known about the fate of guanidine in biological systems. Guanidine has been detected in human urine at concentrations of 7–13 mg L$^{-1}$ (0.12–0.22 mM)[3], but its biosynthetic pathway remains elusive[4]. A recent study also revealed that a variety of microorganisms, including *E. coli*, produce guanidine through unknown mechanisms under nutrient-poor growth conditions, suggesting that guanidine metabolism is biologically significant and is prevalent in natural environments[5].

While nonenzymatic decomposition of guanidine under physiological conditions is very slow[6], soil microbes are able to degrade guanidine using heretofore unknown metabolic pathways[7]. Recently, it was reported that a wide range of microorganisms possesses a class of guanidine riboswitches that control the expression of downstream genes, a majority of which encode proteins involved in nitrogen metabolism, nitrate/sulfate/bicarbonate transport, and guanidine export[5,8–12]. A previously annotated urea carboxylase was reported to carboxylate guanidine to form carboxyguanidine[5], which is degraded by a carboxyguanidine deiminase followed by further degradation by allophanate hydrolase[13]. Another class of enzymes regulated by guanidine riboswitches are annotated as agmatinases in the arginase superfamily[5,9,14,15], which catalyze the breaking of C–N bonds in the guanidyl moiety of agmatine, releasing urea[16]. There is no current explanation for why these enzymes evolved regulation in response to free guanidine.

To date, the only known enzyme that produces guanidine is the ethylene-forming enzyme (EFE) that catalyzes the formation of ethylene and guanidine simultaneously from α-ketoglutarate (AKG) and arginine[17]. Due to biotechnological interests in developing an alternative pathway for renewable production of ethylene, which is the most highly produced organic compound in the petro-chemical industry, the *efe* gene from *Pseudomonas syringae* (a plant pathogen) has been introduced into a variety of microbial species[17]. Some hosts, e.g., *Pseudomonas putida* KT2440 and the cyanobacterium *Synechocystis* sp. PCC 6803 (hereafter *Synechocystis* 6803), have been able to accommodate stable, high-level expression of EFE and thereby sustain enhanced production of ethylene[17–22]. Other species, such as cyanobacterium *Synechococcus elongatus* PCC 7942 (hereafter *Synechococcus* 7942) and *Synechococcus elongatus* PCC 11801 (hereafter *Synechococcus* 11801), however, have not been able to tolerate the high-level expression of EFE, and the recombinant strains suffered severe growth inhibition[23–25] and genome instability. More specifically, spontaneous mutations were found at various sites within the EFE open reading frame in the recombinant *Synechococcus* 7942 strain, which eventually abolished the expression of functional EFE and rescued the growth inhibition[23,24].

In this study, we report a guanidine-degrading enzyme discovered through comparative analysis of multiple cyanobacterial species. We show that guanidine possesses significant toxicity to cyanobacterial cells and destabilizes their genome in response to recombinant EFE expression. *Synechocystis* 6803 is able to degrade and utilize guanidine as a nitrogen source through the activity of an enzyme encoded by the gene *sll1077*, which was previously annotated as an agmatinase in the arginase superfamily. We posit that Sll1077 is more likely a guanidinase because it degrades guanidine rather than agmatine into urea and ammonium. This result is consistent with the finding that there is a conserved sequence motif of the guanidine riboswitch upstream of the *sll1077* ORF in the genome of the wild-type *Synechocystis* 6803 strain. *Synechococcus* 7942 lacks a homologous enzyme in its genome and is unable to mitigate guanidine toxicity. We find that heterologous expression of Sll1077 in a recombinant *Synechococcus* 7942 strain confers the ability to degrade guanidine into nontoxic urea. Co-expression of Sll1077 and EFE in *Synechococcus* 7942 stabilizes the genome of the resultant strain and leads to sustained production of ethylene from light and $CO_2$.

## Results

**Varied guanidine-degradation abilities of cyanobacterial species.** Given that the impacts of guanidine on microorganisms are unclear, we studied guanidine degradability and toxicity in two model cyanobacterial species: *Synechocystis* 6803 and *Synechococcus* 7942. In our preliminary experiments with *Synechocystis* 6803, when nitrate was gradually replaced with guanidine in the culture medium, the guanidine concentrations declined over a period of four days in all cases under photoautotrophic cultivation conditions (Supplementary Fig. 1). In order to rule out the possibility of photochemical degradation, *Synechocystis* 6803 cells were resuspended in the nitrate-deprived mBG11 medium with or without 5 mM guanidine (detailed in "Methods"). In parallel, *Synechococcus* 7942 and heat-killed *Synechocystis* 6803 cells were also resuspended in the nitrate-deprived culture medium supplemented with 5 mM guanidine. We found that *Synechocystis* 6803 cells grown in nitrate-deprived medium exhibited the expected chlorosis phenotype, a process involving degradation of photosynthesis-related pigments to recycle nitrogen for biomass production[26]. As a result, the cultures were still able to double the amount of biomass. On the other hand, cells grown in the guanidine-supplemented medium were able to maintain their green pigmentation and reached a higher cell density after 6 days of photoautotrophic cultivation (Fig. 1a, b). Noticeably, the *Synechocystis* 6803 cells exposed to exogenous guanidine had a slower growth rate than those not exposed to guanidine during the first day, probably due to the toxicity of guanidine (Fig. 1a, b). By contrast, the cultures inoculated with heat-killed *Synechocystis* 6803 or live *Synechococcus* 7942 cells showed a continuous decline of biomass over the period of 6 days (Fig. 1a, b). While the guanidine content in the culture with *Synechococcus* 7942 or heat-killed *Synechocystis* cells did not decline, the continuous increase of biomass in the culture of live *Synechocystis* 6803 cells coincided with a steady decrease of the guanidine concentration in the culture medium (Fig. 1c). To this end, we hypothesized that a guanidine-degrading metabolic pathway may exist in *Synechocystis* 6803 but not in *Synechococcus* 7942.

**Sll1077 confers guanidine degradation in *Synechocystis* 6803.** A comparative proteomic study of the wild-type *Synechocystis* 6803 and the guanidine-producing (*efe*-expressing) strain, JU547[27], showed that the expression of Sll1077, a putative agmatinase, increased by 20-fold in strain JU547 compared to that in the wild-type *Synechocystis* 6803 (Supplementary Data 1). Agmatinase cleaves the C–N bond within the guanidyl moiety of agmatine, which releases putrescine and urea[28]. Interestingly, the expression of *sll1077* is predicted to be under the control of a guanidine riboswitch based on analysis of the RNA sequence upstream of its ORF (Fig. 2a, b)[5]. We hypothesized that Sll1077 might be involved in the metabolism of guanidine in *Synechocystis* 6803 (Fig. 2c). Knockout of *sll1077* in *Synechocystis* 6803, leading to strain PB805W (Δ*sll1077*), did not have any apparent physiological effects on the cells under normal growth conditions (Supplementary Fig. 2), or under nitrate-deprived conditions (Fig. 2d, e). Nevertheless, under nitrate-deprived guanidine-supplemented culture conditions, the cell growth of Δ*sll1077* was severely inhibited compared to the wild-type *Synechocystis* 6803 and the degradation of the light-harvesting components, i.e., phycobilisomes (absorbance at 630 nm) and chlorophyll *a* (absorbance at 680 nm), in Δ*sll1077* was remarkably retarded compared to the

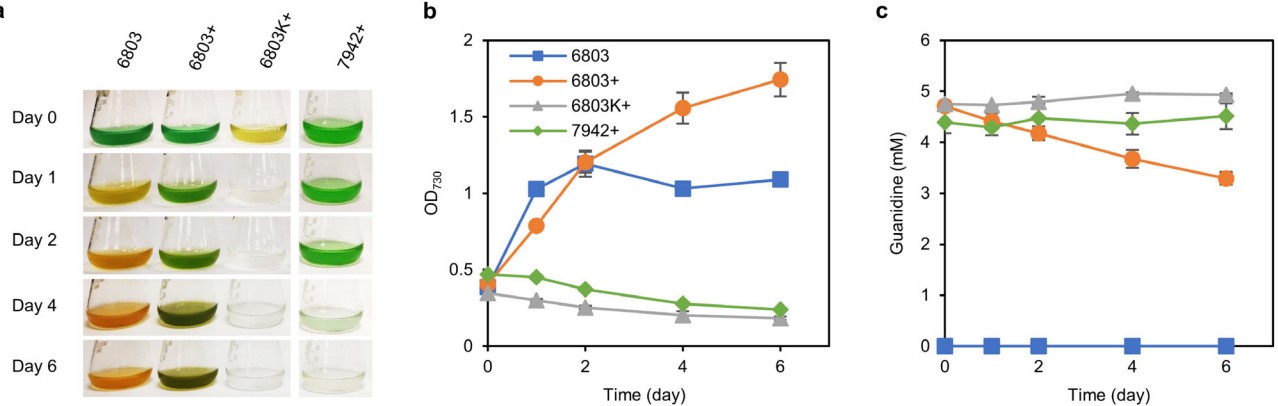

**Fig. 1 Varied capabilities in degrading guanidine between two model cyanobacterial species. a** Phenotypes of *Synechocystis* 6803 and *Synechococcus* 7942 grown in nitrate-deprived medium with or without guanidine. 6803, *Synechocystis* 6803 strain grown in the nitrate-deprived medium; 6803 +, *Synechocystis* 6803 strain grown in nitrate-deprived medium supplemented with 5 mM guanidine. 6803 K +, *Synechocystis* 6803 cells initially killed by heating at 95 °C for 10 min and then resuspended in the nitrate-deprived medium supplemented with 5 mM guanidine. 7942 +, *Synechococcus* 7942 strain grown in nitrate-deprived medium supplemented with 5 mM guanidine. **b** Time courses of cell mass accumulation as monitored by absorbance at 730 nm (OD$_{730}$). **c** Time courses of guanidine concentrations in the culture supernatants. Data represent means and standard deviations from three biological replicates. Source data underlying panels (**b**, **c**) are provided as a Source Data file.

wild-type *Synechocystis* 6803 or Δ*sll1077* cultivated in the medium without guanidine (Fig. 2d–f). Further analysis revealed that the guanidine-degradation capability was abolished in the *Synechocystis* Δ*sll1077* strain (Fig. 2g), a phenotype similar to that of wild-type *Synechococcus* 7942 (Fig. 1a). In addition, during the first day, the biomass of strain Δ*sll1077* incubated with guanidine increased to a much less extent relative to other parallel cases; in the next few days, the biomass of strain Δ*sll1077* incubated with guanidine underwent an autolysis process and the light-harvesting complex gradually deteriorated (Fig. 2d–f).

Overexpression of *sll1077* in *Synechocystis* 6803 was achieved through optimizing the ribosome-binding site (RBS) at the 5'-region as well as tailoring the 3'-region of the expression cassette (Fig. 3a). Among the six tested RBSs, RBSv309 in strain PB809W rendered the strongest expression level (Fig. 3b and Supplementary Fig. 3). While removal of the XhoI restriction site between the *sll1077* and the 6xHis tag sequence at the 3'-region in strain PB812W did not have any apparent effect on the *sll1077* expression level, adding the *rrn*BT1T2 terminator (from *E. coli*) to the 3'-region significantly improved the expression of *sll1077* in PB816W (Fig. 3a, b). Strain PB816W was able to degrade guanidine at a rate ~80% faster than the wild-type *Synechocystis* 6803, which led to a faster cell growth rate in nitrate-deprived medium supplemented with guanidine (Fig. 3c, d). Interestingly, although removal of the 6xHis tag sequence from the 3' end of *sll1077* did not affect the protein expression level (Fig. 3b and Supplementary Fig. 3), it increased the guanidine-degradation rate by about two times and substantially increased the cell growth rate of PB817W (Fig. 3c, d), suggesting that the C-terminus 6xHis tag negatively impairs the guanidine-degrading enzyme activity of Sll1077.

In order to verify that guanidine is degraded by Sll1077 to form urea, according to the enzymatic mechanism of the agmatinase/arginase superfamily[28], Sll1077-His was purified from the crude cell lysate of *Synechocystis* strain PB816W (Figs. 3b and 4a and Supplementary Fig. 4; note that Sll1077 in strain PB817W does not have a His tag for purifying the protein from the cell lysate). Purified Sll1077-His showed an apparent molecular weight of ~45 kDa which is consistent with the predicted molecular weight of 43.8 kDa (Fig. 4a). Incubation of purified Sll1077-His with guanidine at 30 °C resulted in hydrolysis of guanidine and release of urea and ammonium (Fig. 4b–e). We further calculated the

enzyme kinetic parameters, $V_{max} = 0.055 \pm 0.014$ U mg$^{-1}$, $K_M = 5.3 \pm 1.2$ mM, $k_{cat} = 0.040 \pm 0.010$ s$^{-1}$, and $k_{cat}/K_M = 0.0076 \pm 0.0006$ s$^{-1}$ mM$^{-1}$ (Fig. 4f, g). Considering that the His tag inhibits the enzyme activity of Sll1077 by about 2.5-fold under the examined experimental condition (Fig. 3c and Supplementary Fig. 5), the $k_{cat}$ for Sll1077 without a His tag is expected to be higher than that shown in Fig. 4g. It is noteworthy that no cofactors, such as ATP or NAD(P)H, are required to drive the guanidine hydrolysis enzymatic activity of Sll1077, and the presence of ATP has no significant effect on the enzymatic activity of Sll1077 (Supplementary Fig. 6). Therefore, the hydrolytic nature of Sll1077 is more energy-efficient compared to the previously reported guanidine carboxylation pathway[5,13] (Fig. 4h).

**Expression of *sll1077* improves cell tolerance to guanidine**. In order to examine if expressing a recombinant enzyme, Sll1077 from *Synechocystis* 6803, could endow the guanidine-degradation capability in a host strain that does not naturally degrade guanidine, we expressed *sll1077* in *Synechococcus* 7942, resulting in strain GD7942 (+*sll1077*). While the cell growth of *Synechococcus* 7942 was already inhibited by guanidine at concentrations as low as 0.3 mM and was severely inhibited by 1 mM guanidine under photoautotrophic conditions (Fig. 5a), the sll1077-expressing strain GD7942 gained significant tolerance to exogenous guanidine. Particularly, the cell growth of strain GD7942 was not noticeably repressed by as much as 1 mM guanidine present in the culture medium and was only slightly inhibited by 2 mM guanidine (Fig. 5a). We further examined the fate of the exogenous guanidine in the culture medium containing 1 mM guanidine. As expected, while no guanidine degradation occurred in the culture of wild-type *Synechococcus* 7942, the guanidine added into the culture medium of the GD7942 strain was completely degraded over 4 days of photoautotrophic cultivation (Fig. 5b). Since the wild-type *Synechococcus* 7942 does not have any urea biosynthesis or degradation pathways[29], it was expected that urea would be accumulated in the GD7942 culture. Indeed, along with the degradation of guanidine, urea gradually accumulated in the culture supernatants to concentrations of about 1 mM by end of day 4 (Fig. 5b), which is consistent with the pathway annotation[29] and enzymatic reaction stoichiometry (Fig. 4h). We further found that supplementing 5 mM urea into the culture medium of

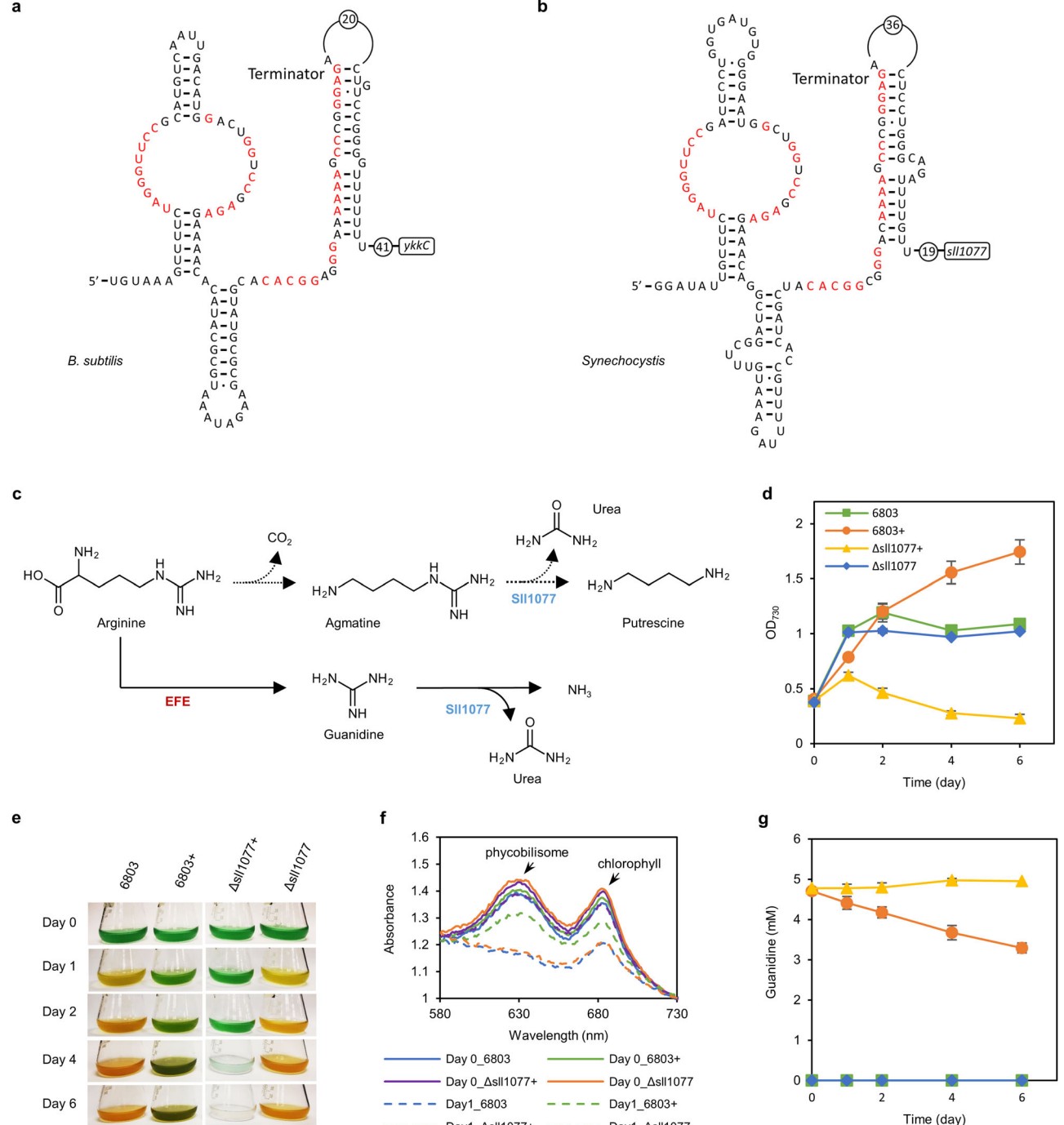

**Fig. 2 Gene *sll1077* is responsible for guanidine degradation in *Synechocystis* 6803. a** The secondary RNA structure of the guanidine riboswitch upstream of the guanidine exporter encoded by the *ykkC* gene in *Bacillus subtilis*. Nucleotides in red are >97% conserved in type I guanidine riboswitch[5]. **b** The secondary RNA structure of predicted guanidine riboswitch upstream of the *sll1077* gene in *Synechocystis* 6803. The consensus guanidine riboswitch nucleotides are depicted in red. **c** The proposed metabolic role of Sll1077 in degrading guanidine as depicted by solid arrows. Sll1077 was previously annotated as an agmatinase in one arginine degradation pathway as depicted by the dotted arrows[15]. **d** Time courses of cell densities of *Synechocystis* 6803 and the Δ*sll1077* strain (PB805W) grown in nitrate-deprived medium with or without guanidine supplementation. In total, 5 mM guanidine was added into the nitrate-deprived medium as indicated by "+" following the strain names. **e** Phenotypes of *Synechocystis* 6803 and the Δ*sll1077* strain (PB805W). **f** Absorbance spectra of cultures at day 0 and day 1 as shown in **d**. Absorbance was normalized to absorbance at 730 nm. **g** Time courses of guanidine concentrations in the culture supernatants. Data represent means and standard deviations from three biological replicates. Source data underlying panels (**d**, **g**) are provided as a Source Data file.

*Synechococcus* 7942 did not show any apparent impact on the cell growth under either nitrate-deprived or nitrate-containing culture conditions (Supplementary Fig. 7), suggesting that *Synechococcus* 7942 is highly tolerant to urea.

**Sll1077 prefers guanidine rather than agmatine as the substrate.** To examine the substrate preference of Sll1077 toward guanidine and agmatine, the crude cell extract of *Synechococcus* 7942 and GD7942 was incubated with 5 mM of either guanidine

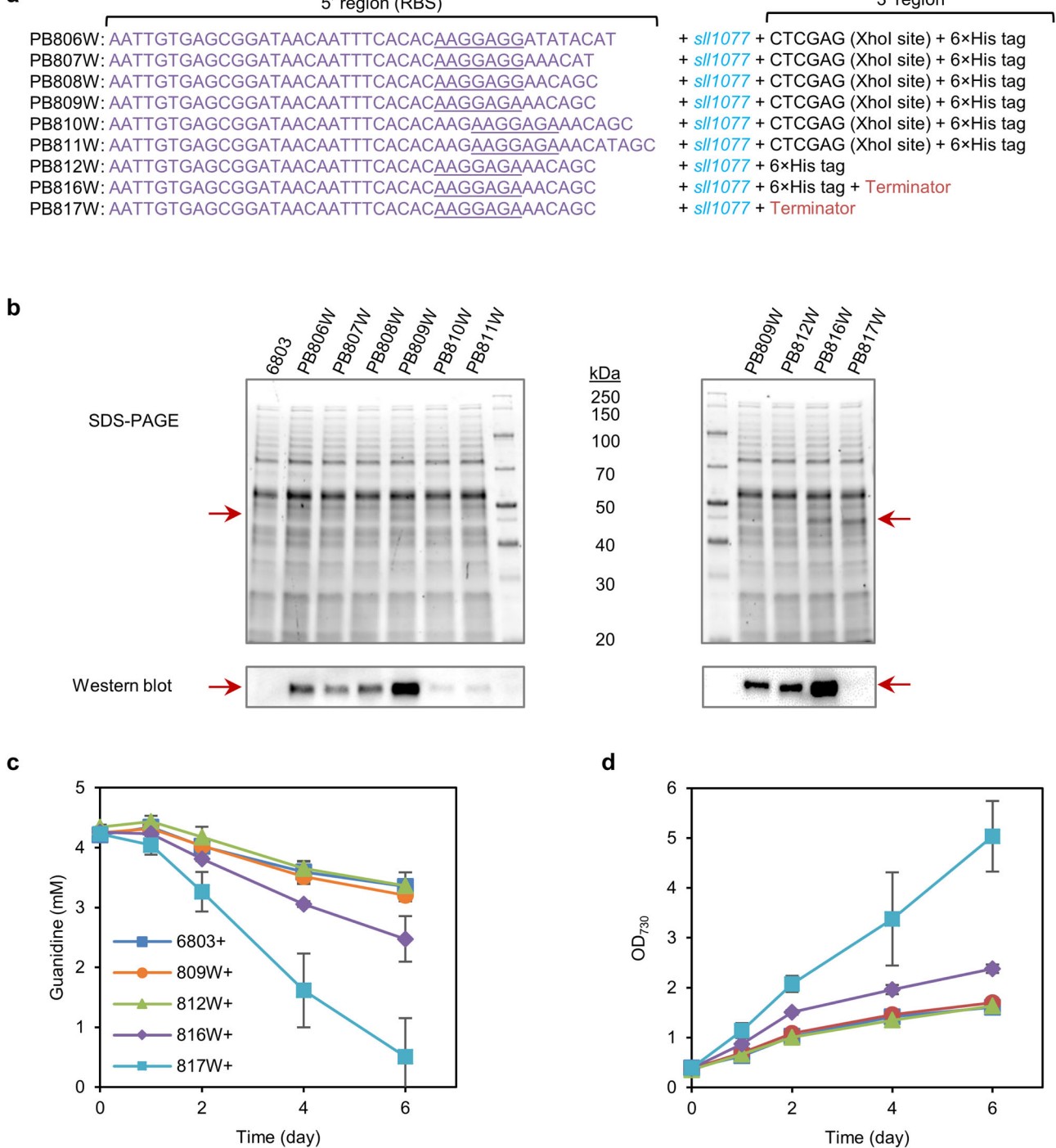

**Fig. 3 Overexpression of Sll1077 accelerates guanidine degradation and promotes biomass growth in *Synechocystis* 6803. a** Strategies for enhancing the overexpression of gene *sll1077* in *Synechocystis* 6803. Gene *sll1077* was overexpressed driven by the *tac* promoter, with its RBS (purple nucleotides, including the entire 5'-untranslated sequence upstream of the start codon of gene *sll1077*) at the 5'-region and the His tag and terminator at the 3'-region optimized. **b** SDS-PAGE and western blotting (His tag) showing the improved expression of Sll1077 in *Synechocystis*. Representative data from two independent experiments. **c** Guanidine-degradation profiles of *Synechocystis* 6803 and *sll1077*-overexpressing strains. **d** Cell growth curves for *Synechocystis* 6803 and *sll1077*-overexpressing strains, indicated by readings of $OD_{730}$ of cell cultures. Data represent means and standard deviations from three biological replicates. Source data underlying panels (**c**, **d**) are provided as a Source Data file.

or agmatine at 30 °C. Surprisingly, we found that the crude cell extract of GD7942 was able to degrade guanidine but not agmatine. The concentration of guanidine incubated with the GD7942 cell lysate decreased by about 2 mM, and concomitantly ~2 mM urea was produced in the reaction mix over the examined 12 h time period (Fig. 5c, d). In contrast, neither guanidine was

consumed nor urea was produced in any other examined cases (Fig. 5c, d). We, therefore, propose denominating Sll1077 as a guanidinase instead of an agmatinase.

**Co-expression of Sll1077 and EFE enhances genomic stability.** Given that the EFE reaction produces not only ethylene but also

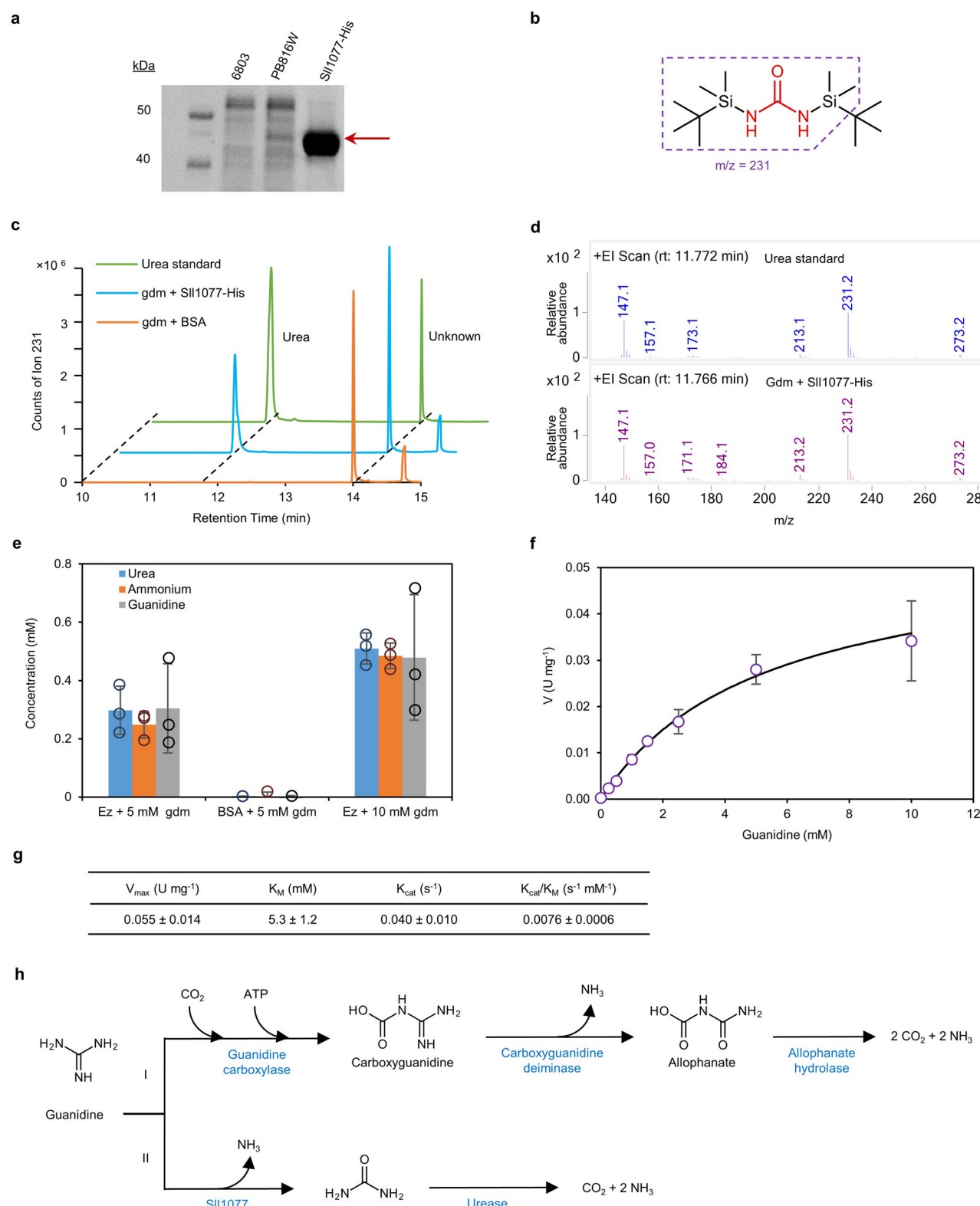

toxic guanidine, which might be responsible for the genomic instability observed upon expression of EFE alone in *Synechococcus* 7942[23,24], we examined if co-expressing Sll1077 and EFE in the *Synechococcus* 7942 host strain would render a stable genome and thereby sustained production of ethylene. We found that following the genetic transformation of *Synechococcus* 7942 and colony restreaking on BG11 agar plates, the recombinant *efe*-

expressing strain, EFE7942, grew considerably slower than wild-type and the initially formed colonies appeared yellow-greenish; subsequently, large and dark-green colonies grew up on the background of the smaller colonies (Fig. 6a). Cultivation of these large and small colonies in the liquid culture revealed that cells from the small colonies, but not from the large ones, retained photosynthetic ethylene productivity. Subsequent colony PCR

**Fig. 4 Confirmation of the guanidine-degrading enzyme activity of Sll1077 through an in vitro enzyme activity assay. a** SDS-PAGE showing the cell extract from *Synechocystis* 6803, PB816W, and purified Sll1077-His. Representative data from two independent experiments. **b** TBDMS derivative of urea. Red font indicates the urea backbone. The boxed portion indicates the main ion detected by GC-MS. **c** Ion counts of ion 231 for TBDMS derivative of urea standard or the TBDMS derivative of the product of reaction mix incubating 5 mM guanidine with either Sll1077-His ("gdm + Sll1077-His") or bovine serum albumin ("gdm + BSA") for 3 h. **d** Mass spectra of the peaks at 11.77 min in **c**. **e** Consumed guanidine and produced urea and ammonium from the reaction mix containing guanidine. "Ez" stands for Sll1077-His enzyme. The reaction was allowed to proceed for 3 h. **f** Plot of the velocities of Sll1077-His versus the concentration of guanidine. U, $\mu$mol min$^{-1}$. **g** Calculated enzymatic properties of Sll1077-His. Since the His tag inhibits the enzyme activity of Sll1077 by about 2.5-fold under the examined experimental condition (Fig. 3c and Supplementary Fig. S9), the $k_{cat}$ for Sll1077 without a His tag is expected to be higher than the value shown herein. **h** Guanidine-degradation pathways identified to date. Pathway I was reported previously, and pathway II is demonstrated in the current study. Data in (**c**, **d**) are representative results from three independent biological replicates, and data in (**e–g**) represent means and standard deviations from three independent biological replicates. Source data underlying panels **e–g** are provided as a Source Data file.

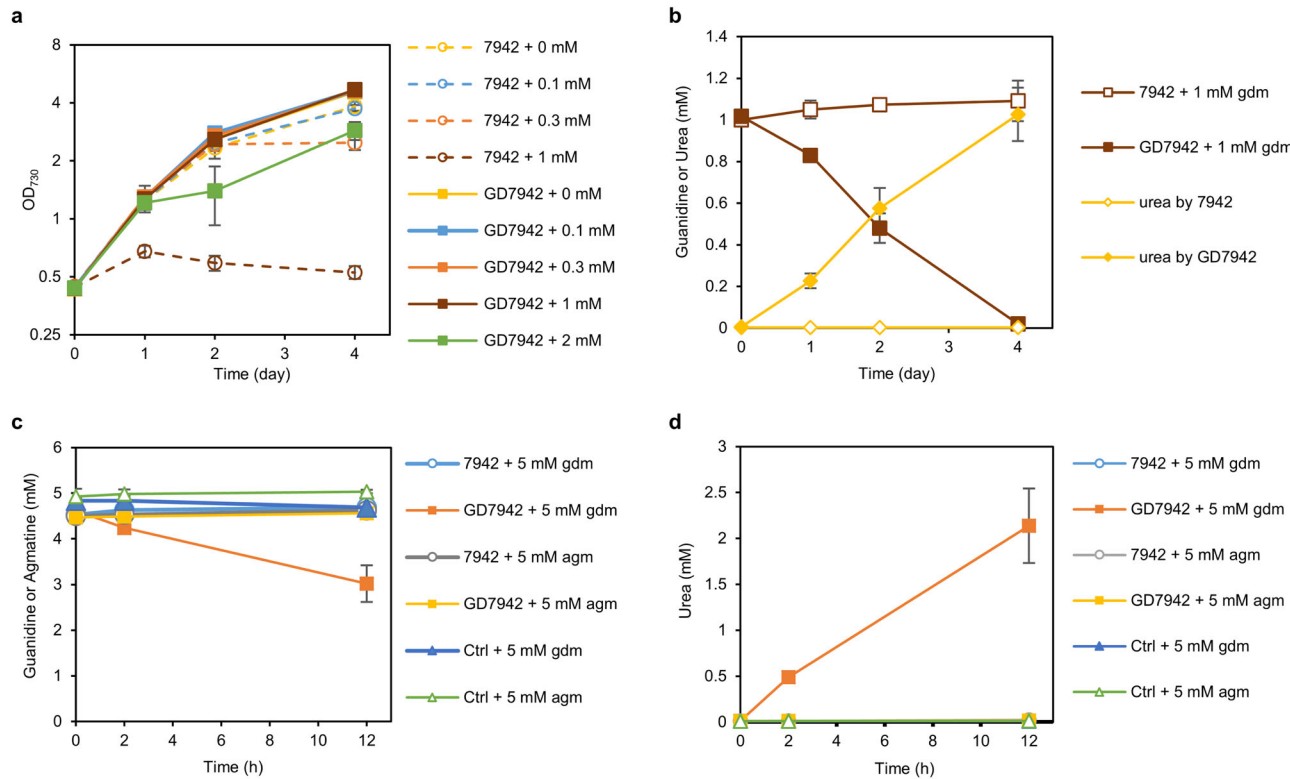

**Fig. 5 Expression of *sll1077* improves the tolerance of *Synechococcus* 7942 to guanidine. a** Cell growth curves for the *Synechococcus* 7942 and GD7942 ( +*sll1077*) grown with various concentrations of exogenous guanidine in the mBG11 medium supplemented with 50 mM bicarbonate. **b** Time courses of guanidine and urea concentrations in the culture supernatants of *Synechococcus* 7942 and GD7942 grown with 1 mM guanidine. **c** Concentrations of guanidine or agmatine in the enzymatic reaction mix along the time course. Either 5 mM guanidine (gdm) or agmatine (agm) was incubated with cell extract of *Synechococcus* 7942 or GD7942. Control (Ctrl) included all the components but the cell extract. **d** Production of urea by the cell extract of *Synechococcus* 7942 or GD7942 incubated with 5 mM guanidine (gdm) or agmatine (agm). All measured urea concentrations were close to zero except for 'GD7942 + 5 mM gdm'. Data represent means and standard deviations from three biological replicates. Source data are provided as a Source Data file.

and DNA sequencing results confirmed that cells from the small colonies retained the correct EFE expression cassette on their genomes, whereas the large colonies consisted of cells with mutations around the EFE expression cassette which abolished expression of EFE (Supplementary Fig. 8). It is noteworthy that restreaking single small colonies onto fresh mBG11-agar plates supplemented with spectinomycin repeatedly resulted in a mixture of large and small colonies after 1–2 weeks of incubation at 30 °C, indicating a constant selective pressure caused by the expression of EFE. In contrast, co-expression of Sll1077 with EFE in *Synechococcus* strain GD-EFE7942 resulted in uniform colony sizes on agar plates at 30 °C (Fig. 6a), and colony PCR and DNA sequencing confirmed that these cells were able to maintain the intact EFE expression cassette on their genome (Supplementary Fig. 8), indicating relief of the selective pressure caused by the expression of EFE. Because EFE exhibits the highest enzyme

activity in the temperature range of 20–25 °C and becomes unstable at temperature above 30 °C[30,31], we decided to routinely maintain strain EFE7942 at 35 °C to suppress the EFE activity and thereby prevent spontaneous mutations from occurring.

The wild-type *Synechococcus* 7942 strain and the *efe*-expressing strains EFE7942 and GD-EFE7942 were then compared in regard to their cell growth rates and ethylene productivities in liquid cultures at 30 °C under photoautotrophic culture conditions. Initially, strain EFE7942 grew considerably slower than the wild-type *Synechococcus* 7942 strain, but gradually grew faster after subsequent re-inoculations, reaching a growth rate similar to that of the wild-type by day 13. In contrast, the GD-EFE7942 strain exhibited a slightly slower growth rate compared to the wild-type strain throughout the entire 13-day cultivation period (Fig. 6b). In terms of the ethylene production, during the first 9 days, strain GD-EFE7942 showed 3–6 times higher volumetric ethylene

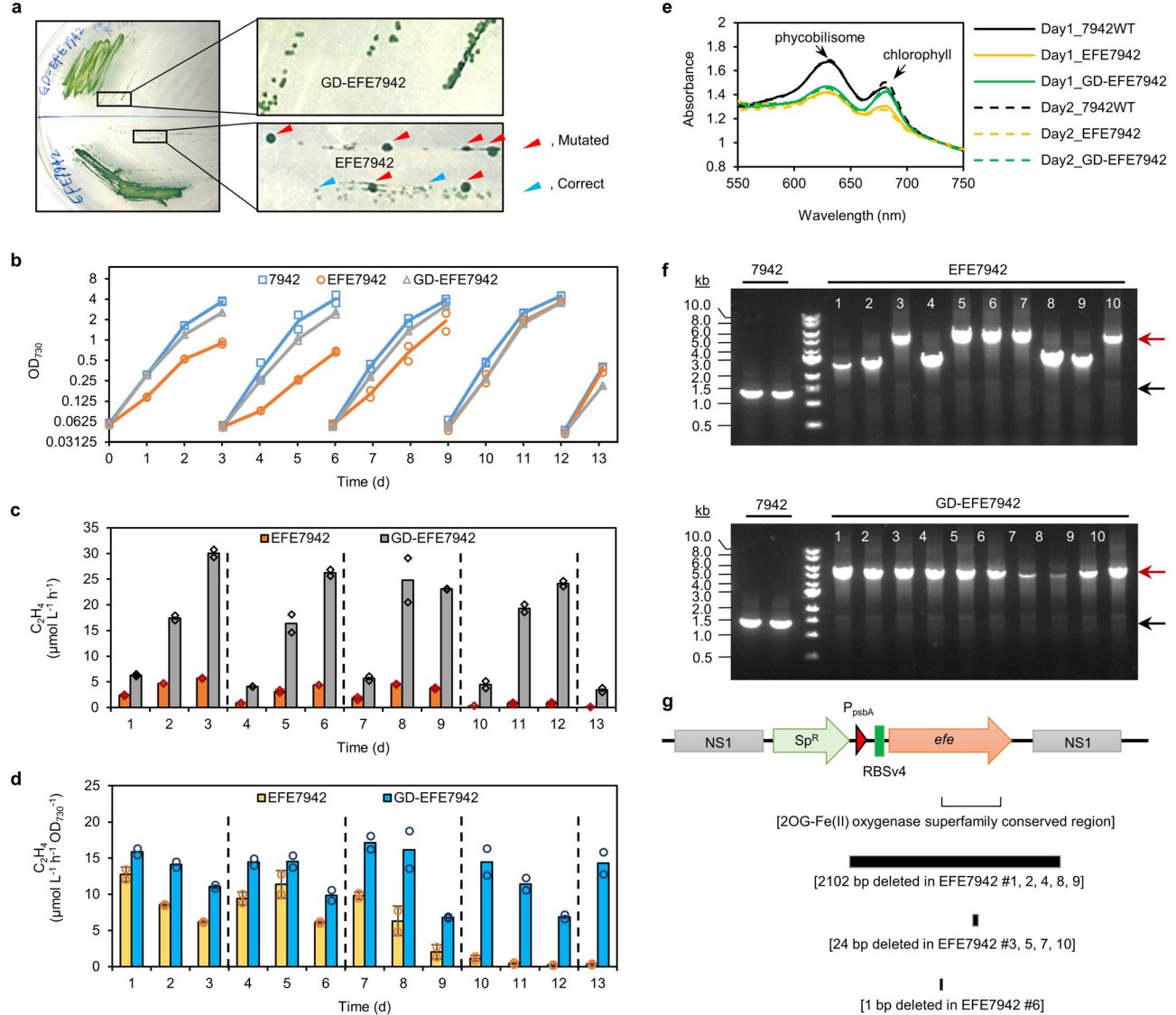

**Fig. 6 Expression of *sll1077* in *Synechococcus* 7942 supports sustained high-level ethylene production. a** Colonies of strains EFE7942 and GD-EFE7942 formed on agar plates at 30 °C. DNA sequencing results revealed that for strain EFE7942, the smaller colonies indicated by cyan triangles harbored the correct EFE expression cassette, while the bigger colonies denoted by red triangles harbored mutated EFE expression cassettes; for strain GD-EFE7942, colony sizes were uniform and DNA sequencing identified no mutations around the EFE expression cassette. **b** Cell growth curves in liquid cultures at 30 °C. Cultures were re-inoculated into fresh BG11 medium (supplemented with 10 mM HEPES, pH 8.2, and 20 mM bicarbonate) media every 3 days. **c** Volumetric ethylene productivities of strains EFE7942 and GD-EFE7942. Data represent means of two independent biological replicates and are overlaid with individual data points. **d** Specific ethylene productivities of strains EFE7942 and GD-EFE7942. Cultures were re-inoculated every three days. Data represent means and standard deviations from two biological replicates. Data represent means of two independent biological replicates and are overlaid with individual data points. **e** Absorbance spectra of cultures shown in (**b–d**) at day 1 and day 2. Absorbance was normalized to absorbance at 730 nm. **f** Two colonies of the wild-type strain 7942, ten colonies of strain EFE7942, and ten colonies of strain GD-EFE7942 were randomly picked from plates spread with diluted day-13 cultures shown in (**b–d**) and were subjected to colony PCR using primers flanking the *efe*-insertion site on the genome. Red arrows indicate the expected PCR product size for strains EFE7942 and GD-EFE7942; black arrows indicate the expected PCR product size for the wild-type 7942 strain. Representative data from two independent experiments. **g** DNA sequencing of the PCR products obtained in (**f**) revealed mutations of the EFE expression cassettes on the genomes of all ten randomly picked EFE7942 colonies, whereas no mutations arose within the genomic region of the EFE expression cassette in any of the ten GD-EFE7942 colonies (Supplementary Data 2). Source data underlying panels (**b–d**) are provided as a Source Data file.

productivities compared to strain EFE7942, with more substantial differences occurring at relatively high cell densities when guanidine accumulated to the highest levels in the culture medium (Fig. 6c and Supplementary Fig. 9). The higher volumetric ethylene productivity of GD-EFE7942 relative to EFE7942 was largely due to the improved cell growth rate and thereby higher cell density (Fig. 6b), yet was also attributed to the

improved specific ethylene productivity (Fig. 6d). During the first 7 days, the specific ethylene productivity of GD-EFE7942 was 1.2–1.8 times higher than EFE7942. The difference increased to 2.6 times by day 8, and to 3.3 times by day 9 (Fig. 6d). Starting from day 10, both the volumetric and specific ethylene productivities of strain EFE7942 dropped substantially and declined to almost zero by day 13 (Fig. 6c, d). The guanidine

production in the EFE7942 culture also started to drop significantly by day 10 (Supplementary Fig. 9). Absorbance spectra of the three examined cultures revealed that the abundance of phycobilisome and chlorophyll a in EFE7942 declined significantly compared to those in the wild-type strain. Although the phycobilisome level remained low in GD-EFE7942 relative to that of the wild-type strain, expression of Sll1077 restored the amount of chlorophyll a in GD-EFE7942 to a level similar to that in the wild-type strain (Fig. 6e), suggesting that degradation of guanidine partially alleviated the stress underlying loss of photosynthesis-related pigments. Further cell growth phenotyping and DNA sequencing analyses revealed that after 13 days of cultivation, approximately half the cells in the EFE7942 culture lost the entire EFE expression cassette, and the other half had DNA mutations on the genome that caused early termination of translation of EFE (Fig. 6f, g, Supplementary Fig. 10 and Supplementary Data 2). We, therefore, conclude that the initial slow growth of EFE7942 was due to the accumulation of toxic guanidine as a byproduct of making ethylene (Supplementary Fig. 9), which provided selective pressure for the cells to acquire mutations that inactivate the EFE gene and eventually rescue the growth inhibition (Fig. 6). By contrast, the GD-EFE7942 strain exhibited consistent cell growth profiles and ethylene productivities during the five consecutive batch cultures (Fig. 6b–d), owing to its engineered capability to mitigate guanidine via Sll1077 (Fig. 5b and Supplementary Fig. 9) and thus enhance genomic stability (Fig. 6f). The slightly slower growth of GD-EFE7942 strain relative to the wild-type Synechococcus 7942 strain is likely attributable to the burden of expression of EFE and Sll1077 as well as redirecting metabolic fluxes from biomass synthesis to ethylene formation. In addition, the ethylene productivity of GD-EFE7942 is comparable to that of the previously engineered high-level-efe-expressing Synechocystis strains, e.g., strain PB752 in our previous work[32], under the examined photoautotrophic culture conditions (Supplementary Fig. 11).

## Discussion

Through comparative analysis of cyanobacterial strains, we were able to identify a guanidine-degrading enzyme, Sll1077, which breaks down guanidine to form urea and ammonium (Figs. 2c, 4, and 5). Sll1077 constitutes a guanidine-degradation pathway that does not require ATP, and is completely different from the recently identified guanidine carboxylation pathway[5,13] (Fig. 4h). Guanidine carboxylase catalyzes the carboxylation of guanidine using ATP as the driving force. However, the product compound carboxyguanidine is unstable and is readily hydrolyzed to form guanidine and $CO_2$ in water, forming an ATP-consuming futile cycle[13]. The efficiency of the guanidine carboxylation pathway largely depends on the rate of removal of carboxyguanidine by the carboxyguanidine deiminase which converts carboxyguanidine to ammonium and allophanate[13]. In contrast, the guanidine-degrading enzyme Sll1077 investigated in this study acts as a deiminase and is able to, without consuming ATP, directly convert guanidine to urea (Fig. 4b–e and Supplementary Fig. 6), which could be further degraded into $CO_2$ and ammonium by the urease in most cyanobacterial species, including Synechocystis 6803[29] (Fig. 4h). Therefore, the Sll1077-associated guanidine-degradation pathway seems more energy-efficient compared to the guanidine carboxylation pathway. The catalytic efficiency of Sll1077, as calculated in the current study (Fig. 4g), seems much lower than some high-efficiency enzymes, such as the agmatinases in E. coli and in Anabaena[33]. It explains why guanidine was accumulated in and out of efe-expressing cyanobacterial cells despite expression of the guanidinase Sll1077 (Supplementary Fig. 9)[27]. However, we realize that the low catalytic efficiency of

guanidinase estimated in the current study is consistent with the fact that biodegradation of guanidine in nature is such a slow process that guanidine has been commonly utilized as a slow-release fertilizer in agriculture[2,7].

Sll1077 represents a class of guanidine-degrading (i.e., guanidinase) enzymes. It was recently uncovered that the main arginine utilization pathway in cyanobacteria is the arginine dihydrolase-mediated route[34,35]. Besides, cyanobacteria employ the agmatinase-mediated pathway for biosynthesis of polyamines[33]. Both Sll1077 and Sll0228 in Synechocystis 6803 have been annotated as putative agmatinases since they both have the conserved regions of the agmatinase/arginase superfamily proteins[14,15]. However, their protein sequences show <25% identities (Supplementary Fig. 12), and we found through protein BLAST that Sll0228 (44% positive hit) is more similar than Sll1077 (37% positive hit) to the biochemically characterized cyanobacterial agmatinase from Anabaena[33]. A recent study also showed that deletion of sll0228 rather than sll1077 significantly impairs the utilization of arginine in Synechocystis 6803[35], which is consistent with a previous report that the agmatinase activity in Synechocystis 6803 is mostly attributed to Sll0228[36]. Despite the enigmatic role of Sll1077, from a bioinformatics approach it was found that the expression of Sll1077 and its analogs (previously annotated to encode agmatinase or arginase enzymes) in a wide range of microorganisms is under the control of guanidine riboswitches (Fig. 2a, b and Supplementary Data 3)[5,8,9]. These genes often form operons with other genes, such as hypA, hypB, SsuA_fam (sll1080 in Synechocystis 6803), TM_PBP2 (sll1081 in Synechocystis 6803), and ABC_NrtD_SsuB (sll1082 in Synechocystis 6803) (Supplementary Data 3)[5]. The expression levels of these genes were all enhanced in guanidine-producing Synechocystis strains compared to wild-type controls according to our proteomic data (Supplementary Data 1) and results from a previous transcriptomic study[37], which is consistent with the modulation mechanism of guanidine riboswitches[5]. Taken together, these results suggest that Sll1077 may be evolved for a function that is completely different from the degradation of arginine or agmatine. Our findings that Sll1077 is able to degrade guanidine and that it prefers guanidine rather than agmatine as the substrate is consistent with the prediction that its expression is under the control of a guanidine riboswitch (Figs. 2a, b and 5c, d)[5], which suggests that Sll1077 and possibly its analogs are evolved for the degradation of guanidine. It is likely that guanidine, formed either biologically or abiotically, is present in the natural environment where Synechocystis 6803 lives, and possessing a guanidinase-encoding gene might have provided a survival advantage to this species. Running a protein BLAST for the Sll1077 peptide sequence (https://blast.ncbi.nlm.nih.gov/) returned over a thousand hits with >50% sequence identities, all of which have been annotated as arginase/agmatinase family proteins (Supplementary Data 4). Whether these proteins possess the capability to degrade guanidine needs to be studied in the future.

Guanidine causes a disorder of pigment metabolism in cyanobacterial cells. Guanidine is known to interact with the peptide backbone and side chains of amino acids, and serves as a protein denaturant when applied at high concentrations (2–6 M)[1,38,39]. At concentrations insufficient to completely unravel the protein structure, guanidine could also be detrimental to biomacromolecules. For example, relatively small amounts of guanidine could trigger unfolding of the active site of ribonuclease A and thereby inactivate the enzyme activity and facilitate the proteolysis process[40]. Another example is that millimolar guanidine could significantly inhibit ammonium nitrification in the nitrifying bacteria in soil[41]. In our study, the presence of guanidine in the culture medium, either from exogenous or endogenous sources, severely inhibited cell growth of wild-type Synechococcus 7942

and the *Synechocystis* Δ*sll1077* strain (Figs. 1a, b, 2d, e, 5a, and 6a, b). These guanidine-sensitive strains exhibited remarkably slow degradation of their light-harvesting components under nitrate-deprived and guanidine-supplemented culture conditions (Figs. 1a and 2e, f). Under nitrogen-poor culture conditions, cyanobacterial cells typically undergo a chlorosis process that involves degrading their phycobiliproteins and chlorophyll as a nitrogen source to support cell growth while simultaneously downregulating photosynthesis in order to reduce the generation of damaging oxygen radicals[26]. Impaired cell growth and retarded pigment degradation in both cultures of *Synechocystis* 6803+ and Δ*sll1077*+ on day 1 (Fig. 2d–f) suggested that induction of nitrogen chlorosis was disrupted by guanidine under the examined culture conditions. Furthermore, the biosynthesis of phycobiliproteins and chlorophyll was severely inhibited in strain *Synechococcus* EFE7942, whereas the biosynthesis of chlorophyll was restored through heterologous expression of Sll1077 in strain GD-EFE7942 (Fig. 6e), which provided additional evidence that guanidine hampers the biosynthesis and remodeling of photosynthesis-related pigments in cyanobacteria.

While the wild-type *Synechococcus* 7942 is sensitive to guanidine and fails to accommodate the high-level expression of EFE (Figs. 1a, b, 5a, and 6), our discovery of the guanidine-degrading activity of Sll1077 was leveraged to generate a derivative strain of *Synechococcus* 7942 that exhibits enhanced genomic stability and stable high-level production of ethylene in prolonged culture, which has not been achieved in prior studies (Fig. 6 and Supplementary Fig. 8)[23–25,42]. It is noteworthy that co-expression of Sll1077 with EFE substantially attenuates, but does not completely eliminate, the accumulation of guanidine in cultures of the engineered *Synechococcus* GD-EFE7942 strain (Supplementary Fig. 9). Although this seems already sufficient for rendering genomic stability and sustained stable ethylene production in GD-EFE7942 (Fig. 6 and Supplementary Figs. 8 and 10), as well as the *Synechocystis* strain PB752[27], it could be possible to obtain a more efficient guanidine-degrading enzyme, perhaps through directed evolution of Sll1077, in order to achieve faster degradation of guanidine and further reduce its toxicity in the future. In summary, this study has advanced our understanding of the biological routes of guanidine metabolism in nature and has demonstrated a useful approach for enhancing the biosynthesis of ethylene by reducing the toxic byproduct guanidine in engineered microorganisms.

## Methods

**Bacterial strains and growth conditions.** *E. coli* NEB5α (New England BioLabs, MA, USA) served as the microbial host for cloning and maintaining all recombinant plasmids, and was routinely grown in LB medium. *Synechocystis* and *Synechococcus* strains were typically grown in a modified chemically defined BG11 medium (mBG11)[27]. Briefly, one liter of mBG11 medium contains the following components in deionized water: 1.5 g $NaNO_3$, 75 mg $MgSO_4 \cdot 7H_2O$, 36 mg $CaCl_2 \cdot 2H_2O$, 6.6 mg citric acid$\cdot H_2O$, 11.2 mg EDTA$\cdot Na_2$, 20 mg $Na_2CO_3$, 30.5 mg $K_2HPO_4$, 6 mg ferric ammonium citrate, 2.86 mg $H_3BO_3$, 1.81 mg $MnCl_2 \cdot 4H_2O$, 0.22 mg $ZnSO_4 \cdot 7H_2O$, 0.39 mg $Na_2MoO_4 \cdot 2H_2O$, 0.08 mg $CuSO_4 \cdot 5H_2O$, and 0.04 mg $CoCl_2 \cdot 6H_2O$. *N*-tris(hydroxymethyl)methyl-2-aminoethanesulfonic acid (TES, pH adjusted to 8.2 by NaOH) and $NaHCO_3$ were supplemented into BG11 medium to final concentrations of 20 mM and 100 mM, respectively, unless otherwise specified. The medium was filtered through sterile 0.22-μm membranes before use. Cyanobacterial liquid cultures were grown under constant light of about 50 μE m$^{-2}$ s$^{-1}$ on a rotary shaker at 150 rpm and 30 °C in a Percival chamber (Percival Scientific, Inc., IA, USA) aerated with 5% $CO_2$. When cyanobacteria were grown on solid medium, 10 mM TES-NaOH (pH8.2), 3 g L$^{-1}$ thiosulfate and 15 g L$^{-1}$ agar were supplemented to the mBG11 medium, and sterilized by autoclaving at 121 °C for 30 min. When appropriate, antibiotics were added to the solid medium to the following final concentrations: 50 mg L$^{-1}$ for spectinomycin and 7 mg L$^{-1}$ for chloramphenicol, respectively.

**Construction of recombinant plasmids.** All enzymes and cloning kits were purchased from New England Biolabs, MA, USA, unless otherwise specified. Kits for DNA purification were purchased from Qiagen, MD, USA. Plasmid pPB305

was constructed by PCR amplification of the DNA fragments of sll1077U, sll1077D, and gene *cat*, and Gibson Assembly into plasmid pBlueScript II SK (+) which was digested with KpnI and SacI. The DNA fragment containing gene *sll1077* was PCR amplified from the genomic DNA of *Synechocystis* 6803 and inserted between the NdeI and XhoI restriction sites on pET30a(+), so that Sll1077 will be tagged with 6xHis, resulting in plasmid pPB300. pPB306 was constructed by PCR amplifying *sll1077-His* from pPB300 and inserting it between the NdeI and SalI restriction sites on pSCPTH (Wang, 2013) using Gibson Assembly Kit. pPB306d was constructed by deleting the lac promoter region on the pBluescript vector backbone via digesting pPB306 with SacI and SapI restriction enzymes and then blunt-ended and self-ligated using T4 DNA polymerase and Quick DNA ligase. pPB307, pPB308, pPB309, pPB310, and pPB311 were constructed by replacing the RBS in pPB306d using the Site-Directed Mutagenesis Kit. pPB312 was constructed by deleting the CTCGAG (XhoI) nucleotides between the sll1077 coding sequence and the 6xHis tag on plasmid pPB309. pPB316 was constructed by inserting the *rrnBT1T2* terminator (from *E. coli* NEB5α) downstream of *sll1077* on pPB312. pPB312 was digested with SalI, dephosphorylated, and then assembled with the terminator *rrnBT1T2* using Gibson Assembly Kit. The map and DNA sequence of pPB316 are included in Supplementary Information. pPB313 was constructed by deleting the 6xHis tag and CTCGAG (XhoI) between the sll1077 coding sequence and stop codon TAA of pPB309. pPB317 was constructed by inserting *rrnBT1T2* downstream of *sll1077* on pPB313, which was digested with SalI and dephosphorylated, using the Gibson Assembly Kit. Plasmid To express the *efe* gene in *Synechococcus* 7942, 1.43 kb of the BbvCI/XhoI fragment containing *psbAp::efe*-FLAG from pJU158 was blunt-ended and ligated to the SmaI site of the neutral site 1 vector pAM1303, resulting in pEFE-FLAG-NS1. To overexpress the *sll1077* gene in *Synechococcus* 7942, 1.69 kb of the BamHI/SalI fragment harboring the *sll1077* expression cassette from pPB317 was cloned into the BamHI/SalI site of a neutral site 4-targeting vector pCX0104-LuxAB-FT[43] to generate pGD7942-NS4. The DNA sequence of genes of interest was all confirmed by DNA sequencing. All plasmids used in this study are listed in Supplementary Table 1 and DNA sequences of representative recombinant DNA constructs are included in Supplementary Data 5. Primers used in constructing all plasmids are detailed in Supplementary Table 2.

**Genome engineering of cyanobacteria.** Transformation of *Synechocystis* was accomplished via natural transformation[44]. Briefly, the wild-type *Synechocystis* 6803 strain was grown in mBG11 medium until the OD$_{730}$ reached ~0.4. Then, 2.5 mL of the culture was condensed to about 0.2 mL via centrifugation and resuspension with the same culture medium. Cells were transferred into a 1.5-mL Eppendorf tube and mixed with 1–2 μg DNA of integration plasmid. The sample was incubated under low light for about 5 h, and mixed once in the middle of the incubation. Cells were then spread onto BG11 plates supplemented with appropriate antibiotics and incubated at 30 °C under about 25 μE m$^{-2}$ s$^{-1}$. Colonies usually appear within 1–2 weeks. Strains PB805W— PB812W, PB816W, and PB817W were constructed by transforming wild-type *Synechocystis* 6803 with integration plasmids pPB305—pPB312, pPB316, and pPB317. Transformation of *Synechococcus* 7942 was completed following a natural transformation protocol[45]. Briefly, 5–10 mL of log-phase cyanobacterial cells were centrifuged at 3000 × g for 5 min. Then, the cell pellet was washed with 10 mL of 10 mM NaCl followed by centrifugation and disposal of the supernatant. Cells were resuspended with 0.3 mL of mBG11 medium and were mixed with 1 μL of purified DNA (about 0.1–0.3 μg), followed by incubation with gentle shaking in darkness overnight. 0.1 mL of transformation mix (diluted when necessary) was spread onto BG11 plates supplemented with appropriate antibiotics and incubated at 30 °C under about 25 μE m$^{-2}$ s$^{-1}$. Colonies typically appear within 1–2 weeks. Transformation of *Synechococcus* 7942 with integration plasmids pEFE-FLAG-NS1 or pGD7942-NS4 resulted in strain EFE7942 and GD7942, respectively. The *efe* expression cassette was PCR amplified from the genomic DNA of EFE7942 strain using primers NS15 and NS16, and inserted into the neutral site 1 of the genome of *Synechococcus* GD7942, resulting in strain GD-EFE7942. The complete segregation of genomes was verified via colony PCR, followed by DNA sequencing of the PCR products amplified using primers (listed in Supplementary Table 2) flanking the modified regions of the cyanobacterial genomes. DNA sequencing was performed by GENEWIZ (South Plainfield, NJ, USA). The strains used in this study are listed in Supplementary Table 1.

**SDS-PAGE and western blotting.** When the OD$_{730}$ of cyanobacterial culture reached 0.5–1.0, ~5 OD$_{730}$ mL (i.e., 10 mL if the OD$_{730}$ of the culture equals 0.5) of cells were collected via centrifugation at 3220 × g, 24 °C for 5 min and removal of supernatants. The cell pellets were stored at −80 °C until use. Upon running SDS-PAGE, cells were resuspended with 0.5 mL of cold 0.1 M potassium phosphate buffer (pH 7.0) supplemented with DTT (0.2 mM) and Halt Protein Inhibitor Cocktail (Thermo Fisher Scientific, MA, USA), and mixed with 0.2 g of 0.1-mm-diameter acid-washed glass beads, and then subjected to bead-beating at 4 °C for 5 min using the Digital Disruptor Genie (Scientific Industries, Inc., NY, USA). The cell lysate was centrifuged at 4 °C, 18,000 × g for 10 min, and then the supernatant containing soluble proteins was transferred into a new Eppendorf tube placed on ice. The protein concentrations were estimated using the Bradford assay (Thermo Fisher Scientific, MA, USA). Then, 2.5 μg of protein from each sample was mixed

with 2× SDS-PAGE sample buffer (950 μl Bio-Rad 2× Laemmli Sample Buffer +50 μl BME) in a PCR tube and incubated at 99 °C for 5 min using a thermocycler. Samples were then loaded onto Mini-PROTEAN® TGX Stain-Free™ precast gels (Bio-Rad Laboratories, CA, USA), and electrophoresis was conducted at 150 V for about 45 min. Gels were imaged using UV excitation in a FluorChem Q imager (ProteinSimple, CA, USA).

Western blotting was conducted using Pierce™ G2 Fast Blotter (Thermo Fisher Scientific, MA, USA). HisProbe™-HRP Conjugate (Catalog number: 15165, Thermo Fisher Scientific, MA, USA) was used as the probe (at 1:500 dilution) to detect the Sll1077-His. In total, 2 mL SuperSignal™ West Dura Extended Duration Substrate (Catalog number: 34075, Thermo Fisher Scientific, MA, USA) was added to each blot and incubated for 5 min before the chemiluminescent blots were imaged using FluorChem Q imager (ProteinSimple, CA, USA).

**In vitro enzyme activity assay**. His-tagged Sll1077 i.e., Sll1077-His, was first purified from *Synechocystis* PB816W. PB816W was grown in 250 mL mBG11 medium under 120 μE m$^{-2}$ s$^{-1}$ until an OD$_{730}$ of about 2, and then cells were harvested via centrifugation at 4700 × g, 24 °C for 10 min followed by removal of supernatants. The cell pellets were stored at −80 °C. Cells were subsequently resuspended with 10 mL of cold 20 mM Tris · HCl (pH 7.5) containing 1 mM DTT and 1× Halt Protein Inhibitor Cocktail (Thermo Fisher Scientific, MA, USA), and lysed by sonication in an ice-water bath via 100 cycles of 5-s-on-5-s-off. The cell lysate was centrifuged at 4 °C, 17,000 × g for 30 min. Then, the supernatant containing soluble proteins was loaded onto His GraviTrap (GE Healthcare) column to purify Sll1077-His following the user manual. The Sll1077-His was eluted with an elution buffer containing 20 mM Tris · HCl (pH7.5), 500 mM NaCl and 500 mM immidizole. The eluted samples were dialyzed against a dialysis buffer that contains 20 mM Tris · HCl (pH7.5), 50 mM NaCl and 1 mM MnCl$_2$ at room temperature for 2 h and then dialyzed against fresh dialysis buffer at 4 °C over night. The protein concentration was determined using a Bradford assay kit.

Guanidine was added into 1.6 mL of reaction buffer (the same as the above dialysis buffer) to a series of final concentrations, i.e., 0, 0.25, 0.5, 1, 1.5, 2.5, 5, and 10 mM. The reaction mixes were incubated at 30 °C in a water bath for about 15 min before 0.2 mL of purified Sll1077-His (0.67 mg mL$^{-1}$) or BSA (0.67 mg mL$^{-1}$) was added to initiate the reaction. In some experiments, ATP was added into the reaction mix to a final concentration of 1 mM. The reaction mixes were incubated at 30 °C in a water bath for 3 h, and 0.5 mL was sampled at 0, 1, and 3 h. Immediately after being sampled, samples were quenched with 50 μL 2 N HCl, and were neutralized with 50 μL 2 N NaOH after all samples were collected. Samples were stored at −20 °C until further analysis.

For measuring urea in the samples, 150 μL of each sample was mixed with 600 μL methanol, vortexed, and then 150 μL chloroform was added, vortexed, and then 450 μL water was added, vortexed, followed by centrifugation at 17,000 × g for 2 min. The aqueous phase (~1.2 mL) was transferred into a clean Eppendorf tube, air-dried overnight and then lyophilized before being mixed with 35 μL of pyridine and 50 μL of MTBSTFA + 1% TBDMCS (Regis Technologies, Inc., IL, USA). The derivatization reaction mix was incubated at 70 °C for 30 min. A series of concentrations of urea standards were dissolved in the in vitro enzyme assay buffer, lyophilized and derivatized side by side with the enzyme assay samples in order to establish a calibration curve to quantify the urea. The derivatized samples were centrifuged at 17,000 × g, room temperature for 5 min, and then 1 μL the supernatants were analyzed on GC-MS using a method adapted from a previous study[46]. Briefly, GC-MS was performed using an Agilent 7890 gas chromatograph (GC) connected to an Agilent 5977 A mass spectrometer (MS). The GC was equipped with a HP-5ms column (2 × 15 m × 0.25 mm i.d., 0.25-μm film thickness; J&W Scientific, Cat. No. 19091S-431). The injection volume was 1 μL and all samples were run in splitless mode with an inlet temperature of 270 °C. The helium flow rate was set to 1 mL min$^{-1}$. The GC oven temperature was held at 80 °C for 1 min, ramped at 20 °C min$^{-1}$ to 140 °C, and then ramped at 4 °C min$^{-1}$ to 174 °C, and then ramped at 40 °C min$^{-1}$ to 285 °C. A 3-min column backflushing was performed after each run at a post run temperature of 315 °C. Mass spectra were obtained in scan mode over the range 140–595 m/z. MS source temperature was 230 °C, and MS quad temperature was 150 °C. Agilent MassHunter Workstation GC/MS Data Acquisition Version 10.0.368 was used for collecting GC-MS data.

For HPLC quantification of guanidine, 100 μL samples were subjected to methanol/chloroform extraction and air-dried as stated above, and then resuspended with 1 mL of deionized water before being analyzed using the method described below.

For analysis of ammonium, a 300-μL sample was diluted by dH$_2$O to 1.5 mL and then analyzed using an Ammonia Gas Sensing Electrode (Cat. No. 9512BNWP, Thermo Fisher Scientific, MA, USA) according to the manual.

**In vitro substrate preference assay for Sll1077**. *Synechococcus* 7942 and GD7942 strains were grown in 250-mL flasks each containing 60 mL mBG11 medium supplemented with 50 mM NaHCO$_3$ on a rotary shaker at 130 rpm, under 1% CO$_2$, 60 μE m$^{-2}$ s$^{-1}$ until OD$_{730}$ reached about 1.5. Then 60 OD$_{730}$ mL of cells were harvested and centrifuged at 4700 × g, 24 °C for 10 min. The supernatants were discarded, and the cell pellets were kept at −80 °C until use. Subsequently, cell pellets were resuspended with 1 mL 100 mM Tris · HCl (pH 8.0) containing 1 mM DTT and 1× Halt Protein Inhibitor Cocktail (Thermo Fisher Scientific, MA, USA),

and lysed by sonication in an ice-water bath. The lysates were then centrifuged at 4 °C, 17,000 × g for 30 min. Then the cell extract (supernatants) were used for the following in vitro assay: 1.54 mL 100 mM Tris · HCl (pH 8.0), 20 μL MnCl$_2$, 20 μL NH$_4$Cl, 400 μL cell extract, and 20 μL 500 mM guanidine · HCl or agmatine · HCl (with a total reaction volume of 2 mL). All the components but the cell extract in each reaction mix were mixed together and incubated in a 30 °C water bath for about 15 min before the cell extract was added into the reaction mix to start the assay. In the control experiments, cell extracts were replaced by the 400 μL 100 mM Tris · HCl (pH 8.0) containing 1 mM DTT and 1× Halt Protein Inhibitor Cocktail. During incubation, 0.5 mL of sample was taken from the reaction mixes at 0, 2, and 12 h time points, and were immediately mixed with 50 μL 2 N HCl to quench any enzymatic activity. 50 μL 2 N NaOH was then added the samples to neutralize the pH followed by storage at −20 °C. After all samples were collected, 150 μL of each sample was used for quantification of urea using GC-MS and another aliquot of 150 μL was used for quantification of guanidine and agmatine using HPLC. The samples were subsequently processed following a similar protocol as described in the previous section.

**Quantification of guanidine and agmatine using HPLC**. Guanidine was quantified using a protocol modified from a previous method[27]. Briefly, guanidine hydrochloride and agmatine standard solutions and biological samples were passed through 0.2-μm diameter membrane filters and then were analyzed using an Agilent 1200 Series HPLC (Agilent, USA) equipped with a Multi-Wavelength Detector and a set of Dionex IonPac™ CS14 cation-exchange guard (4 mm × 50 mm) and analytical columns (4 × 250 mm; Thermo Fisher Scientific, MA, USA). The column temperature was held at 30 °C. The mobile phase was 20 mM methanesulfonic acid dissolved in 5% acetonitrile in water, and it was pumped through the column at a constant flow rate of 1.0 mL min$^{-1}$ for 30 min. The sample injection volume was 50 μL. Guanidine and agmatine were eluted at around 3.7 min and 7.9 min, respectively, and were monitored by their absorbance at 195 nm using the Agilent ChemStation software Rev. B.04.01 SP1.

**Guanidine-tolerance and -degradation test**. For the guanidine-tolerance test, *Synechococcus* strains were grown in 20 mL mBG11 medium supplemented with 0–1 mM guanidine and 50 mM NaHCO$_3$. For the guanidine-degradation test, *Synechocystis* strains were grown in 10 mL mBG11 free of nitrate while supplemented with 50 mM NaHCO$_3$ and 5 mM or 1 mM guanidine chloride, under constant light of 50 μE m$^{-2}$ s$^{-1}$ on a rotary shaker at 150 rpm and 30 °C. Every day, 1 mL of culture was sampled for reading OD$_{730}$ and then transferred into an Eppendorf tube and centrifuged at 17,000 × g at room temperature for 2 min. The supernatants were stored at −20 °C for later analysis of guanidine.

**Production of ethylene from engineered *Synechococcus* strains**. The *Synechococcus* EFE7942, GD-EFE7942, and WT (a negative control) strains were grown in mBG11 supplemented with 10 mM HEPES-NaOH (pH 8.2) and 20 mM NaHCO$_3$ at 35 °C until OD$_{730}$ reached ~1.0. Subsequently, each strain was inoculated into 50 mL fresh medium with an initial OD$_{730}$ of about 0.05, and grown under continuous light of 100 μE m$^{-2}$ s$^{-1}$ at 30 °C aerated with 1% CO$_2$ at a rate of 50 mL min$^{-1}$. Every day, 2 mL of culture was sampled for ethylene productivity assay, measurement of OD$_{730}$ and guanidine analysis. After every 3 days of cultivation, appropriate volumes of cultures were centrifuged and resuspended with 50 mL fresh medium to an initial OD$_{730}$ of about 0.05.

**Measurement of ethylene produced from cyanobacteria**. One milliliter cyanobacterial culture was transferred into a 17-mL glass test tube, sealed immediately with a rubber stopper, and incubated under 100 μE m$^{-2}$ s$^{-1}$ at 30 °C with shaking. After 3 h of incubation, 250 μL gas was sampled from the headspace of the test tube using a sample-lock syringe and injected into the Shimadzu GC-2010 system equipped with a flame ionization detector (FID) and a RESTEK column (length, 30.0 m; inner diameter, 0.32 mm; film thickness, 5 μm). The GC-FID was operated under the following conditions: carrier gas, helium; inlet temperature, 200 °C; split ratio, 25; inlet total flow, 40.4 mL min$^{-1}$; Pressure, 79 kPa; column flow, 1.53 mL min$^{-1}$; linear velocity, 32.1 cm s$^{-1}$ (Flow Control Mode); purge flow, 0.5 mL min$^{-1}$; column temperature, 130 °C; equilibration time, 2 min; hold time, 2 min; FID temperature, 200 °C; sampling rate, 40 ms; stop time, 2 min; FID makeup gas, He; FID makeup flow, 30 mL min$^{-1}$; H2 flow, 40 mL min$^{-1}$; airflow, 400 mL min$^{-1}$. The standard curve for calibrating ethylene concentrations was generated using the ethylene calibration gas purchased from GASCO (Oldsmar, FL, USA). The GC-FID data collection and analysis were operated using Shimadzu LabSolutions Lite software Release 5.52.

**Shotgun proteomics**. *Synechocystis* 6803 and the ethylene-producing JU547 were inoculated into 3 × 50 mL mBG11 with an initial OD$_{730}$ of 0.1. When OD$_{730}$ reached about 0.5, 60 OD$_{730}$ mL cells were collected via centrifugation at 3220 × g, 4 °C for 5 min. The cell pellets were washed with 25 mL cold wash buffer (50 mM Tris · HCl, pH 8.0 and 10 mM CaCl$_2$) and centrifuged again, followed by washing with 20 mL and 1 mL washing buffer. The supernatants were discarded and cells were frozen at −80 °C. Three biological replicates were included for each strain. Comparative proteomic analyses of *Synechocystis* 6803 and JU547 were conducted

following our previously published method[47]. In brief, cell pellets taken out of −80 °C were lysed by sonication with a program of 12 cycles of 10-sec-on-2-min-off on ice. The supernatants were collected via centrifugation and the protein concentrations were analyzed using Bradford assay (Thermo Scientific, Rockford, IL). Then, 75 μg of total protein for each sample was denatured by incubating with 8 M urea and 5 mM dithiothreitol (DTT) at 37 °C for an hour, followed by protein alkylation with 15 mM iodoacetamide for 30 min in the dark at room temperature. Next, the protein solution was diluted fourfolds with the Tris buffer (50 mM Tris · HCl, pH 8.0, and 10 mM CaCl$_2$), and then digested with Trypsin Gold (Promega, Madison, WI) with an enzyme/protein ratio of 1:100 (w/w) for 18 h at 37 °C. After digestion, peptides were desalted by passing through a Sep-Pak C18 plus column (Waters Corporation, Milford, MA) and eluted with 1 mL of Solvent B (0.1% formic acid, 70% acetonitrile, and 30% water). The peptides were then dried completely in a SpeedVac and redissolved in 100 μL of Solvent A (0.1% formic acid, 5% acetonitrile, and 95% water). The entire peptides were loaded onto an in-house packed biphasic strong cation-exchange (SCX)/reverse-phase capillary column using a pressure cell. Online two-dimensional peptide separation was conducted using a liquid chromatography pump (Surveyor MS Pump Plus, Thermo Scientific, San Jose, CA) by pulsing with eleven increasing concentrations of ammonium acetate for SCX separation, followed by a 2-h reverse-phase gradient program from 100% Solvent A to 60% Solvent B. The peptides were continuously emitted and analyzed in a Thermo Scientific LTQ XL ion trap mass spectrometer (Thermo Scientific, San Jose, CA) operated under the data-dependent acquisition mode. The full mass spectra were collected using the Xcalibur data system (Thermo Scientific, San Jose, CA) in the range of 300–1700 m/z, followed by selecting the top five most abundant ions for the collision-induced dissociation tandem mass (MS/MS) events. To analyze the peptide information, a protein database was generated using the *Synechocystis* sp. proteins obtained from UniProt database and 37 common contaminant proteins. The tandem mass spectra were analyzed using the proteomics search algorithm ProLuCID (version 1.0)[48], followed by peptide screening using DTASelect (version 2.0)[49] with a false discovery rate of 0.05. A minimum of two peptides were used to identify a protein. Comparison of protein levels was conducted using the pipeline software Patternlab for Proteomics (version 4.0.0.38)[50] using the TFold test. Proteins that satisfy both fold change and *p*-value were used to indicate differential expression levels between the compared groups. The raw proteomics data have been deposited to the ProteomeXchange Consortium[51] with the dataset identifier PXD025040.

**Reporting summary**. Further information on research design is available in the Nature Research Reporting Summary linked to this article.

## Data availability
Data supporting the findings of this work are available within the paper and its Supplementary Information files. A reporting summary for this Article is available as a Supplementary Information file. The original proteomics dataset generated in this study has been deposited to the ProteomeXchange Consortium[51] with the dataset identifier PXD025040. The DNA sequencing data confirming the intact or mutated *efe*-insertion region of the cyanobacterial genomes are included in Supplementary Data 2, and have been deposited to GenBank with accession numbers MZ465674, MZ465675, MZ465676, and MZ465677. Source data are provided with this paper.

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

## Acknowledgements

The authors would like to thank Dr. Chi Zhao for his assistance in analyzing ammonium. This work was authored in part by Alliance for Sustainable Energy, LLC, the manager and operator of the National Renewable Energy Laboratory for the U.S. Department of Energy (DOE) under Contract No. DE-AC36-08GO28308. Funding was provided by DOE Office of Energy Efficiency and Renewable Energy BioEnergy Technologies Office (B.W. and J.Yu.). This study was supported in part by the DOE Genomic Science Program under award DE-SC0019404 (B.W., Y.X., C.H.J. and J.D.Y.), DE-SC0019388 (B.W. and J.D.Y), and DE-SC0018344 (B.W. and J.D.Y) and grants from the NIH/NIGMS (R37 GM067152 and R01 GM107434 to C.H.J.) and by DE-AR0000203 (X.W. and J.S.Yuan).

## Author contributions

B.W. and J.Yu. conceived the work. B.W. designed and performed most of the experiments and drafted the manuscript. Y.X. constructed recombinant *Synechococcus elongatus* strains. X.W. and J.Yuan performed proteomic analysis. C.H.J and J.D.Y. performed discussion, critical review, and revision of the manuscript. All authors read and approved the manuscript.

## Competing interests

B.W. and J.Yu were listed as inventors in a provisional patent application (Application No. 63126828), which is about the application of the guanidine-degrading enzyme for detoxifying guanidine and for enhanced bioproduction. The remaining authors declare no competing interests.

## Additional information

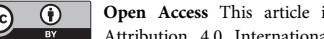

