## [Peer Review File · Nature Communications]

REVIEWER COMMENTS

Reviewer #1 (Remarks to the Author):

The paper by Wang et al describes the ability of the *sll1077* gene product (*Sll1077*) from the unicellular cyanobacterium *Synechocystis* to provide "guanidinase" activity to *Synechocystis* itself or to another cyanobacterium, *Synechococcus elongatus*. The work uses a genetic-physiological approach to convincingly show that *Sll1077* provides *Synechocystis* or *S. elongatus* with the capacity to tolerate toxic guanidine, degrading it and producing urea as a product. This, in turn, allows a better expression of an ethylene-forming enzyme in *S. elongatus*, which is of biotechnological interest. The authors draw as their main conclusion that *Sll1077* is an enzyme that carries out a previously unknown "guanidinase" reaction. There are however some issues, detailed below, that need to be clarified.

1. Lines 164-168: the guanidinase activity of *Sll1077* seems to be very slow, with incubations carried out for up to 12 h, which is unusual in enzymology. This leads to the main weakness of this work. Whereas it is clear that *Sll1077* can degrade free guanidine, it is not clear that this is the primary activity of this enzyme. For the scientific community to be able to evaluate this, it would be necessary that the authors present a basic enzymological characterization, showing K_m , K_{cat} and catalytic efficiency of the enzyme. Having purified the protein expressed in *E. coli*, this characterization should be straightforward. In case the catalytic efficiency were low, perhaps ATP or other cofactors might enhance the activity in contrast to what the authors claim.
2. To complete the stoichiometry of the reaction catalyzed by the enzyme, ammonia released (in addition to urea) should have been determined and shown.
3. Lines 319-322: I'm not sure Zhang et al (ref. 34) described a role for *Sll0228* in arginine utilization. The picture is more complete today. The main arginine utilization enzyme in cyanobacteria is "arginine-guanidine removing enzyme" (*AgrE*) as described for *Anabaena* (PMID: 30636068). (The homologue in *Synechocystis* is *Sll1336*.) Additionally, *Sll0228* is more similar than *Sll1077* to the best characterized cyanobacterial agmatinase, also from *Anabaena* (PMID: 29923645). This supports the proposal that *Sll1077* has a distinct activity, degrading free guanidine as shown here or, alternatively, removing urea from a guanidine group attached to another molecule (evidently not arginine or agmatine). It would be great if this could be clarified.

Minor points:

4. Lines 92-93: you can hardly speak of the chlorotic phenotype of a heat-killed cell suspension. Please rephrase this sentence.
5. Line 113: 10-fold, is it 20-fold in Table S1?
6. Legend to figure 5: c is d and d is c.
7. Strain PCC 7002 is described in M&M but not used in the work described in this manuscript (I think).

Reviewer #2 (Remarks to the Author):

This paper addresses the problem of genetically engineering cyanobacteria so that they produce ethylene, which would represent an advance for renewable energy. Previous work had established an enzyme to synthesize ethylene, but the resulting strain did not grow efficiently and spontaneously lost the gene because it created toxic levels of guanidine. This paper identifies a gene, *sll1077*, that is highly up-regulated in response to ethylene production. While annotated as an agmatinase, the paper shows convincingly that it actually degrades guanidine, a hypothesis that fits into recent work of guanidine riboswitches (are RNAs that bind guanidine and turn on genes in response to high levels). One of the genes commonly regulated by these riboswitches are annotated as agmatinases. Thus, the paper solves the biochemical function of a mysterious gene, addresses the question of why agmatinase genes are associated with guanidine riboswitches and shows how to more efficiently

produce ethylene by using this gene to degrade guanidine.

I enjoyed reading this paper. My background is in the bioinformatics of microbial genetics and RNA, and found the parts of the paper not quite in my area to be pretty accessible. The paper's experiments convinced me of its conclusions, and I liked the mixture of strain optimization and biotechnology with chemistry and detailed microbial genetics.

I would like to emphasize that, although I have written a vast number of comments, they are minor, and mostly directed at helping readers like me more easily understand the paper. The writing is, however, already quite good.

MINOR COMMENTS

The last sentence of Discussion seems to imply that this is the first study in which the biosynthesis of any desired molecule was accelerated by catalysing the removal of a toxic byproduct. I don't know the field that well, but I'm skeptical that no-one has done this with any kind of specific molecules. If in fact no-one has done this, please make the sentence more clear (the phrase beginning with "focusing upon the specific example" makes it sound a bit uncertain if this is completely novel). Otherwise, if there have been previous studies, even if they weren't relate to ethylene or guanidine, please rephrase the sentence to clarify. I still think the paper is interesting.

Why were the 6803 cells grown without nitrate or guanidine ("6803" in Fig. 1b) able to grow at all? Aren't they deprived of a nitrogen source? Do they have some nutrients carried over from before the beginning of the culture? The effect is even greater with 7942+0mM cells in Fig. 5a, where the OD730 increases by a factor of 8.

Please state in what media were the cells grown in Fig. 6b?

In Fig. 4b,c, a derivative of urea was visualized by using ion 231. In the orange curve for the lower graph in Fig. 4c, this curve is showing the product of the reaction of guanidine with sll1077, but not a TBDMS derivative of guanidine, right? So, where does ion 231 come from?

VERY MINOR COMMENTS

It would be helpful to state explicitly that mBG11 is a chemically defined medium and/or that leaving out nitrate means that there's no nitrogen source (if I'm right about that).

The paper states that guanidine riboswitches control genes for "small multidrug resistance (SMR) transporters". As the paper states, these genes were shown to be guanidine transporters, so saying they're SMR transporters is a bit imprecise.

I would have liked a brief description in the Introduction of what specific kind of effects one sees related to genomic instability. Does this just describe mutations in the recombinant EFE gene, or also elsewhere in the genome? It became more clear later in the paper, but at the beginning I found the term "genome instability" a bit vague.

Explain "an expected chlorosis phenotype" on first use, or use the phrase like "maintain their green pigment" (from later in the sentence). I wasn't familiar with the word "chlorosis".

I generally had some problems understanding why the light-harvesting components get degraded. I would have thought that these components get degraded when the cells are sick, but there seemed to have been cases where healthier cells have these components degrade. For example "cell growth of

delta-sll1077 was severely inhibited compared to wild-type *Synechocystis* 6803 and the degradation of the light harvesting components ... in delta-sll1077 was remarkably retarded compared to ... wild-type *Synechocystis* 6803". A bit more explanation would be helpful for readers unfamiliar with cyanobacterial biology. I see that there's some text about this in Discussion, but it would have been helpful to have had this information in Results, so that I know about what the significance is while I'm seeing the results.

Fig. 2f. It'd be nice to mark 630 nm and 680 nm on the X-axis. I missed this information in the text, and had forgotten about it by the time I looked at the figure.

In Fig. 3a, it'd be nice to show the sequence up to the start codon, so that it's clear exactly where this sequence fits in the genome, for people wanting to replicate this. This information could also go in a supplementary figure or table.

Fig. 3b. It'd be convenient to have the red arrow (indicating the sll1077 band) also for the left gel.

"which led to a faster cell growth rate in nitrate-deprived medium"  "which led to a faster cell growth rate in nitrate-deprived medium supplemented with guanidine" (right?)

Based on Fig. 3 and related text, strain PB817W was the best at degrading guanidine, but in the experiments related to Fig. 4, strain PB816W was used. It would have been minorly helpful for me to have pointed out that PB817W lacks the His tag, so it couldn't have been easily used to purify the enzyme.

Fig. 5b could be labeled a bit more clearly. I found it a bit confusing that the codes that "gdm_7942" means that guanidine was measured, whereas in previous graphs the annotation of strain numbers has indicated what nitrogen source was added. Also, it would be nice to write something like "+5 mM guanidine in media" somewhere. Similarly, in Fig. 5c, I assume that whichever molecule is measured on the Y-axis is the same as what was supplemented in the medium, e.g., for 7942+5mM gdm that means that guanidine was supplemented, and the Y-axis gives the concentration of guanidine. This could be stated explicitly.

I think a part of the legend for Fig. 5c and d are mistakenly swapped. The legend for part c says "Production of urea by the cell extract...", but the graph in part c is labeled "Guanidine of Agmatine (mM)". Similarly, part d says in the legend "Concentrations of guanidine or agmatine", but the Y-axis in the graph is labeled for urea.

It'd be nice to state in the legend for part d that all datapoints are very close to zero except for GD7942 + 5 mM gdm, because I can only see the green triangles and the yellow squares. I assume the other 3 experiments are also there, but it'd be nice to have this explicit.

When the paper says "nitrate-replete culture conditions" (page 12, line 207), I think this means minimal media supplemented with guanidine, but in general it sounds like some kind of rich medium. I recommend writing for example "...under either nitrate-deprived culture conditions or with guanidine as the sole nitrogen source".

For the text related to Fig 6b,c, I think it'd be helpful to explicitly state what is going on with EFE7942. It seems to me that the reason that strain EFE7942 is initially slow (Fig 6b) is that it's creating guanidine as a byproduct of making ethylene. It starts to grow faster as time progresses, presumably because the cells acquire mutations that inactivate the EFE gene (supported by Fig 6c).

Why is the growth of GD-EFE7942 slightly slower than the wild-type growth? Because of the energy for expressing EFE and the guanidinase? An explicit statement would be nice.

What is the significance of the information on phycobilisome and chlorophyll a (Fig 6e)?

Check the grammar here: "possessing sll1077 has rendered survival advantage".

Also "It is noteworthy that co-expression of Sll1077 with EFE substantially attenuate"

Reviewer #3 (Remarks to the Author):

The observation and subsequent description of a novel guanidine degrading enzyme that controls genomic stability of ethylene producing strains of cyanobacteria is most innovative. In addition, it is of a wider interest in the scientific community - the fate of guanidine in biological systems. I find the experiments and results presented and discussed convincing, and I like the use of two different cyanobacteria, *Synechocystis* PCC 6803 vs *Synechococcus* PCC 7942, as microbiological chassis. The experimental approach is logical and includes needed controls. Moreover, the important section comparing guanidine with agmatine as substrate for the identified/suggested enzyme Sll1077 is similarly well performed and presented. I agree with the final words of the authors: "..this study advanced our understanding of the biological routes of guanidine metabolism in nature and has demonstrated a new approach for enhancing biosynthesis of target molecule(s) by reducing byproduct(s), focusing upon the specific example of stabilizing ethylene production in engineered microorganisms.

Reply to Reviewers' Comments

Reviewer #1 (Remarks to the Author):

1. The paper by Wang et al describes the ability of the sll1077 gene product (Sll1077) from the unicellular cyanobacterium Synechocystis to provide “guanidinase” activity to Synechocystis itself or to another cyanobacterium, Synechococcus elongatus. The work uses a genetic-physiological approach to convincingly show that Sll1077 provides Synechocystis or S. elongatus with the capacity to tolerate toxic guanidine, degrading it and producing urea as a product. This, in turn, allows a better expression of an ethylene-forming enzyme in S. elongatus, which is of biotechnological interest. The authors draw as their main conclusion that Sll1077 is an enzyme that carries out a previously unknown “guanidinase” reaction. There are however some issues, detailed below, that need to be clarified.

Reply: We thank the referee for her/his time on reviewing our work.

2. Lines 164-168: the guanidinase activity of Sll1077 seems to be very slow, with incubations carried out for up to 12 h, which is unusual in enzymology. This leads to the main weakness of this work. Whereas it is clear that Sll1077 can degrade free guanidine, it is not clear that this is the primary activity of this enzyme. For the scientific community to be able to evaluate this, it would be necessary that the authors present a basic enzymological characterization, showing Km, Kcat and catalytic efficiency of the enzyme. Having purified the protein expressed in E. coli, this characterization should be straightforward.

Reply: We thank the referee for the insight. The original experiment was aimed at qualitatively examining the guanidinase activity of Sll1077, and therefore the assay duration was extended to 12 h in order to allow relatively high amount of urea being accumulated and thereby to facilitate the downstream detection of the product urea. We appreciate the reviewer's suggestion of a more comprehensive enzymological characterization of Sll1077. We have since conducted *in vitro* enzyme assays and monitored concentrations of substrate and products at 0 h, 1 h and 3 h after the start of the enzymatic reaction. In the revised manuscript, we incorporated new results to show the calculated Km, Kcat and catalytic efficiency of the enzyme Sll1077 towards guanidine. Because expression of His-tagged Sll1077, i.e., Sll1077-His, in *E. coli* resulted in formation of inclusion bodies, we purified Sll1077-His from the cell lysate of recombinant cyanobacterium strain *Synechocystis* PB816W, and then performed enzymological assay. The results have been included in Figure 4c-g and in the text (line 173 – 177 in the tracked version) in the revised manuscript. The Kcat (0.04 s^{-1}) and Kcat/Km ($0.0076 \text{ s}^{-1} \text{ mM}^{-1}$) of Sll1077-His are relatively low compared to some high-efficiency enzymes. Since in our original manuscript we found that the His-tag at the C-terminus of the Sll1077-His impairs the guanidine-degrading enzyme activity (Figure 3), we herein further compared the guanidine-degrading enzyme activities from the cell lysates of the wild-type *Synechocystis* 6803 (expressing native basal level Sll1077), recombinant PB816W (expressing additional Sll1077-His) and PB817W (expressing additional Sll1077). We found that removal of the His tag from the C-terminus of Sll1077 enzyme improved the enzyme activity by about 2.5- fold (Figure S9). Based on these results, we expect that the Kcat of the untagged Sll1077 enzyme towards guanidine will be higher than the value shown in Figure 4g. Although removal of His tag from C-terminus of

Sll1077 would improve the catalytic efficiency of this “guanidinase”, it is likely that its catalytic efficiency is still much lower than some high-efficiency enzymes. For example, the agmatinase SpeB in *E. coli* has a K_{cat} of 120 s^{-1} and a K_{cat}/K_m of $81.5 \text{ s}^{-1} \text{ mM}^{-1}$ (Ref 33). Nevertheless, we realize that the estimated low catalytic efficiency of “guanidinase” in the current study is consistent with the fact that biodegradation of guanidine is a slow process in nature which makes guanidine a slow-release fertilizer in agriculture. It is also consistent with the accumulation of guanidine inside and outside of EFE-expressing *Synechocystis* (Ref 27) and *Synechococcus* (Fig. S4) cells despite expression of the guanidinase Sll1077. We discussed this in the text, line 352 – 358. In addition, the expression of Sll1077 is under control of a guanidine riboswitch (Figure 2a,b), and we demonstrated that Sll1077 degrades guanidine rather than agmatine (Figure 5c,d). Taken together, we propose denominating Sll1077 as a “guanidinase” instead of an agmatinase (line 233 – 238 in the tracked version).

3. In case the catalytic efficiency were low, perhaps ATP or other cofactors might enhance the activity in contrast to what the authors claim.

Reply: We confirmed that addition of ATP does not significantly affect the enzyme activity of Sll1077 towards guanidine (Figure S8). (line 179 in the tracked version)

4. To complete the stoichiometry of the reaction catalyzed by the enzyme, ammonia released (in addition to urea) should have been determined and shown.

Reply: We thank the referee for the suggestion. We determined the ammonium concentrations in the reaction mix at the beginning and at the end of the *in vitro* enzyme activity assay, and confirmed that ammonium was produced from the enzyme reaction (Figure 4e). The approximately 1:1:1 molar ratio of consumed guanidine and produced urea and ammonium completed the stoichiometry of the biochemical reaction catalyzed by Sll1077.

5. Lines 319-322: I'm not sure Zhang et al (ref. 34) described a role for Sll0228 in arginine utilization. The picture is more complete today. The main arginine utilization enzyme in cyanobacteria is “arginine-guanidine removing enzyme” (AgrE) as described for Anabaena (PMID: 30636068). (The homologue in Synechocystis is Sll1336.) Additionally, Sll0228 is more similar than Sll1077 to the best characterized cyanobacterial agmatinase, also from Anabaena (PMID: 29923645). This supports the proposal that Sll1077 has a distinct activity, degrading free guanidine as shown here or, alternatively, removing urea from a guanidine group attached to another molecule (evidently not arginine or agmatine). It would be great if this could be clarified.

Reply: We thank the referee for suggesting these two relevant papers. These two papers are now included in our discussion (line 359 – 362).

6. Lines 92-93: you can hardly speak of the chlorotic phenotype of a heat-killed cell suspension. Please rephrase this sentence.

Reply: We thank the referee for the suggestion. We have revised the statement accordingly – line 97.

7. Line 113: 10-fold, is it 20-fold in Table S1?

Reply: We thank the referee for pointing out the discrepancy. We have revised the text accordingly to reflect the results in Table S1 – line 117.

8. *Legend to figure 5: c is d and d is c.*

Reply: We appreciate the referee for pointing out our mistake. We have corrected the legend of Figure 5 accordingly.

9. *Strain PCC 7002 is described in M&M but not used in the work described in this manuscript (I think).*

Reply: We thank the referee for pointing out our mistake. Since strain PCC 7002 is not relevant to the current study, we have removed it from the text (line 446 – 448).

Reviewer #2 (Remarks to the Author):

*1. This paper addresses the problem of genetically engineering cyanobacteria so that they produce ethylene, which would represent an advance for renewable energy. Previous work had established an enzyme to synthesize ethylene, but the resulting strain did not grow efficiently and spontaneously lost the gene because it created toxic levels of guanidine. This paper identifies a gene, *sll1077*, that is highly up-regulated in response to ethylene production. While annotated as an agmatinase, the paper shows convincingly that it actually degrades guanidine, a hypothesis that fits into recent work of guanidine riboswitches (are RNAs that bind guanidine and turn on genes in response to high levels). One of the genes commonly regulated by these riboswitches are annotated as agmatinases. Thus, the paper solves the biochemical function of a mysterious gene, addresses the question of why agmatinase genes are associated with guanidine riboswitches and shows how to more efficiently produce ethylene by using this gene to degrade guanidine.*

Reply: The authors appreciate the referee for the positive comments on our work.

2. I enjoyed reading this paper. My background is in the bioinformatics of microbial genetics and RNA, and found the parts of the paper not quite in my area to be pretty accessible. The paper's experiments convinced me of its conclusions, and I liked the mixture of strain optimization and biotechnology with chemistry and detailed microbial genetics.

Reply: We thank the referee for her/his time on reviewing our work.

3. I would like to emphasize that, although I have written a vast number of comments, they are minor, and mostly directed at helping readers like me more easily understand the paper. The writing is, however, already quite good.

Reply: Thank you for providing these very helpful suggestions on improving our manuscript.

4. *The last sentence of Discussion seems to imply that this is the first study in which the biosynthesis of any desired molecule was accelerated by catalysing the removal of a toxic byproduct. I don't know the field that well, but I'm skeptical that no-one has done this with any kind of specific molecules. If in fact no-one has done this, please make the sentence more clear (the phrase beginning with "focusing upon the specific example" makes it sound a bit uncertain if this is completely novel). Otherwise, if there have been previous studies, even if they weren't relate to ethylene or guanidine, please rephrase the sentence to clarify. I still think the paper is interesting.*

Reply: We have revised our manuscript to make the contribution of our work more clear (line 424 – 428 in the tracked version).

5. *Why were the 6803 cells grown without nitrate or guanidine ("6803" in Fig. 1b) able to grow at all? Aren't they deprived of a nitrogen source? Do they have some nutrients carried over from before the beginning of the culture?*

Reply: Cyanobacteria typically undertake a chlorosis process when nitrogen sources are not available in the culture medium. The process involves a series of actions. The cells would first degrade intracellular nitrogen storage compounds, such as cyanophycin (multi-L-arginyl-poly[L-aspartic acid]) in *Synechocystis* 6803. Next, they degrade their pigments (protein complexes) that form their photosynthesis antennas, called phycobilisomes. The regenerated nitrogen is able to support roughly one more round of cell division as you can see for the case of “6803” in Figure 1b, but the process would also result in a yellowish color of the culture due to loss of photosynthesis-related pigments (case “6803” in Figure 1a). We have discussed this in the text (line 88 –91, 403 – 406 in the tracked version).

6. *The effect is even greater with 7942+0mM cells in Fig. 5a, where the OD730 increases by a factor of 8.*

Reply: Thank you for pointing out this mistake. The 7942+0mM culture in Figure 5a actually contains nitrate in the medium. We simply added various concentrations of guanidine into the culture to examine its toxicity to the cells. We have revised the legend of Figure 5 to reflect the correct medium we used for this experiment.

7. *Please state in what media were the cells grown in Fig. 6b?*

Reply: The culture medium used in experiments that led to Figure 6b was mBG11 supplemented with 10 mM HEPES (pH8.2) and 20 mM bicarbonate. We have included this medium recipe in the legend of Figure 6. The medium recipe is also described in the section of “Production of ethylene from engineered *Synechococcus* strains” in Materials and Methods (line 623 – 630 in the tracked version). For your information, folks traditionally use TES to buffer the BG11 medium for growing *Synechocystis* 6803, while typically use HEPES to buffer the BG11 medium for cultivating *Synechococcus* elongatus 7942. To our best knowledge, the molecular structures of these two buffering compounds are similar, and there is no obvious difference between the buffering capacities of these two compounds.

8. *In Fig. 4b,c, a derivative of urea was visualized by using ion 231. In the orange curve for the lower graph in Fig. 4c, this curve is showing the product of the reaction of guanidine with sll1077, but not a TBDMS derivative of guanidine, right? So, where does ion 231 come from?*

Reply: The figure has been updated in the revised manuscript in order to show the new results from a 3-h (rather than 12-h in the original manuscript) duration enzyme assay. Urea is not readily detectable using GC-MS and has to be derivatized in order to be detectable (Figure 4b). In the new Figure 4c, “Urea standard” stands for TBDMS-derivatized urea standard (a positive control); “gdm + Sll1077-His” stands for TBDMS-derivatized samples from the reaction mix containing guanidine and Sll1077-His enzyme; “gdm + BSA” stands for TBDMS-derivatized samples from the reaction mix containing guanidine and BSA (a protein unable to degrade guanidine and therefore unable to produce urea; a negative control). Figure 4c indicates that a compound that has a retention time same as that of urea was detected in the reaction mix of “gdm + Sll1077-His”, but not in the negative control “gdm + BSA”. Mass spectra in Figure 4d show that the compound with the retention time of 11.77 min from “gdm + Sll1077-His” in Figure 4c matches urea. Using both retention time and mass spectra, we were able to confirm that urea was produced by incubating guanidine with Sll1077-His enzyme.

9. *It would be helpful to state explicitly that mBG11 is a chemically defined medium and/or that leaving out nitrate means that there's no nitrogen source (if I'm right about that).*

Reply: To avoid confusion, we have explicitly described the composition of the mBG11 medium in the Materials and Methods section. (Line 433 – 440)

10. *The paper states that guanidine riboswitches control genes for "small multidrug resistance (SMR) transporters". As the paper states, these genes were shown to be guanidine transporters, so saying they're SMR transporters is a bit imprecise.*

Reply: The authors agree that it is imprecise to use “SMR” here to refer to these genes since it was recently demonstrated that guanidine riboswitch-controlled “SMRs” are actually guanidine transporters (Ref 12). We have revised the text to refer to these genes as guanidine exporters rather than SMRs (Line 41).

11. *I would have liked a brief description in the Introduction of what specific kind of effects one sees related to genomic instability. Does this just describe mutations in the recombinant EFE gene, or also elsewhere in the genome? It became more clear later in the paper, but at the beginning I found the term "genome instability" a bit vague.*

Reply: Thank you for the suggestion. We have added more description in the Introduction to clarify where the mutations occurred on the genome (Line 60 – 63).

12. *Explain "an expected chlorosis phenotype" on first use, or use the phrase like "maintain their green pigment" (from later in the sentence). I wasn't familiar with the word "chlorosis".*

Reply: We have defined the term “chlorosis” in the text (Line 89 – 90, 403 – 406).

13. *I generally had some problems understanding why the light-harvesting components get*

degraded. I would have thought that these components get degraded when the cells are sick, but there seemed to have been cases where healthier cells have these components degrade. For example "cell growth of delta-sll1077 was severely inhibited compared to wild-type Synechocystis 6803 and the degradation of the light harvesting components ... in delta-sll1077 was remarkably retarded compared to ... wild-type Synechocystis 6803". A bit more explanation would be helpful for readers unfamiliar with cyanobacterial biology. I see that there's some text about this in Discussion, but it would have been helpful to have had this information in Results, so that I know about what the significance is while I'm seeing the results.

Reply: Thank you for the suggestion. We have defined the term “chlorosis” in the Results section, which explains the role of pigment degradation upon nitrogen starvation. We have added “We found that while Synechocystis 6803 cells grown in nitrate-deprived medium exhibited the expected chlorosis phenotype, a process involving degradation of photosynthesis-related pigments to recycle nitrogen for biomass production. As a result, the cultures exhibited an expected chlorosis phenotype and were still able to double the amount of biomass.” and “Under nitrogen-poor culture conditions, cyanobacterial cells typically undergo a chlorosis process that involves degrading their phycobiliproteins and chlorophyll as a nitrogen source to support cell growth while simultaneously downregulating photosynthesis in order to reduce the generation of damaging oxygen radicals.” (Line 89 – 90, 403 – 406).

14. Fig. 2f. It'd be nice to mark 630 nm and 680 nm on the X-axis. I missed this information in the text, and had forgotten about it by the time I looked at the figure.

Reply: Thank you for the suggestion. We have added labels “phycobilisome” and “chlorophyll” in Figure 2f and 6e.

15. In Fig. 3a, it'd be nice to show the sequence up to the start codon, so that it's clear exactly where this sequence fits in the genome, for people wanting to replicate this. This information could also go in a supplementary figure or table.

Reply: The sequence of “5’ region (RBS)” includes DNA sequence up to the start codon of gene sll1077. An explanation is now added to the figure legend. The DNA sequence and map of plasmid pPB316, which was used to construct strain PB816W, is also included in the Supplementary Information.

16. Fig. 3b. It'd be convenient to have the red arrow (indicating the sll1077 band) also for the left gel.

Reply: Red arrows indicating the *sll1077* band are now added for the left gel.

17. "which led to a faster cell growth rate in nitrate-deprived medium"  "which led to a faster cell growth rate in nitrate-deprived medium supplemented with guanidine" (right?)

Reply: The reviewer is correct. We have revised the sentence accordingly to make it more clear (Line 144 – 145).

18. Based on Fig. 3 and related text, strain PB817W was the best at degrading guanidine, but in the experiments related to Fig. 4, strain PB816W was used. It would have been minorly helpful

for me to have pointed out that PB817W lacks the His tag, so it couldn't have been easily used to purify the enzyme.

Reply: We have revised the sentence accordingly to clarify this point (Line 169 –170).

19. *Fig. 5b could be labeled a bit more clearly. I found it a bit confusing that the codes that "gdm_7942" means that guanidine was measured, whereas in previous graphs the annotation of strain numbers has indicated what nitrogen source was added. Also, it would be nice to write something like "+5 mM guanidine in media" somewhere. Similarly, in Fig. 5c, I assume that whichever molecule is measured on the Y-axis is the same as what was supplemented in the medium, e.g., for 7942+5mM gdm that means that guanidine was supplemented, and the Y-axis gives the concentration of guanidine. This could be stated explicitly.*

Reply: We have revised the legend next to the graphic panel of Figure 5b to avoid confusion. For example, “gdm_7942” is replaced with “7942 + 1 mM gdm”. We have also added the sentence “In contrast, neither guanidine was consumed nor urea was produced in any other examined cases (Fig. 5c,d).” in the main text (line 236 – 237) to explicitly describe our results shown in Figure 5c,d.

20. *I think a part of the legend for Fig. 5c and d are mistakenly swapped. The legend for part c says "Production of urea by the cell extract...", but the graph in part c is labeled "Guanidine of Agmatine (mM)". Similarly, part d says in the legend "Concentrations of guanidine or agmatine", but the Y-axis in the graph is labeled for urea.*

Reply: We thank the reviewer for pointing out this mistake. We have corrected the figure legends for Figure 5c,d accordingly.

21. *It'd be nice to state in the legend for part d that all datapoints are very close to zero except for GD7942 + 5 mM gdm, because I can only see the green triangles and the yellow squares. I assume the other 3 experiments are also there, but it'd be nice to have this explicit.*

Reply: We thank the reviewer for the suggestion. We have added “All measured urea concentrations were close to zero except for ‘GD7942 + 5 mM gdm’” to the legend of Figure 5d.

22. *When the paper says "nitrate-replete culture conditions" (page 12, line 207), I think this means minimal media supplemented with guanidine, but in general it sounds like some kind of rich medium. I recommend writing for example "...under either nitrate-deprived culture conditions or with guanidine as the sole nitrogen source".*

Reply: “nitrate-replete” meant medium containing nitrate. To avoid confusion, we have replaced “nitrate-replete” with “nitrate-containing”. (Line 228)

23. *For the text related to Fig 6b,c, I think it'd be helpful to explicitly state what is going on with EFE7942. It seems to me that the reason that strain EFE7942 is initially slow (Fig 6b) is that it's creating guanidine as a byproduct of making ethylene. It starts to grow faster as time progresses, presumably because the cells acquire mutations that inactivate the EFE gene (supported by Fig 6c).*

Reply: We thank the reviewer for the suggestion. We have added the following sentence to the text related to Figure 6. “We therefore conclude that the initial slow growth of EFE7942 was due to accumulation of toxic guanidine as a byproduct of making ethylene (Fig. S4), which provided selective pressure for the cells to acquire mutations that inactivate the EFE gene and eventually rescued the growth inhibition (Fig. 6).” (Line 304 – 307)

24. Why is the growth of GD-EFE7942 slightly slower than the wild-type growth? Because of the energy for expressing EFE and the guanidinase? An explicit statement would be nice.

Reply: We thank the reviewer for the suggestion. We have added the following sentence to the text related to Figure 6b. “The slightly slower growth of GD-EFE7942 strain relative to the wild-type *Synechococcus* 7942 strain was likely attributed to the burden of expression of EFE and Sll1077 as well as redirecting metabolic fluxes from biomass synthesis to ethylene formation.” (Line 310 – 313)

25. What is the significance of the information on phycobilisome and chlorophyll a (Fig 6e)?

Reply: We discussed the related results in the Discussion section (line 403 – 413). In this revised manuscript, we further added the following sentence to the text related to Figure 6e in the Results section. “mitigation of guanidine might have alleviated the stress on biosynthesis of photosynthesis-related pigments.” (Line 296 – 301)

26. Check the grammar here: "possessing sll1077 has rendered survival advantage".

Reply: We have rephrased the sentence to “possessing a guanidinase-encoding gene might have provided a survival advantage to this species”. (Line 387 – 388)

27. Also "It is noteworthy that co-expression of Sll1077 with EFE substantially attenuate"

Reply: Thank you for pointing out our mistake. We have revised the sentence to “It is noteworthy that co-expression of Sll1077 with EFE substantially attenuates...”. (Line 418 – 419)

Reviewer #3 (Remarks to the Author):

The observation and subsequent description of a novel guanidine degrading enzyme that controls genomic stability of ethylene producing strains of cyanobacteria is most innovative. In addition, it is of a wider interest in the scientific community - the fate of guanidine in biological systems. I find the experiments and results presented and discussed convincing, and I like the use of two different cyanobacteria, Synechocystis PCC 6803 vs Synechococcus PCC 7942, as microbiological chassis. The experimental approach is logical and includes needed controls. Moreover, the important section comparing guanidine with agmatine as substrate for the identified/suggested enzyme Sll1077 is similarly well performed and presented. I agree with the final words of the authors: "...this study advanced our understanding of the biological routes of guanidine metabolism in nature and has demonstrated a new approach for enhancing

biosynthesis of target molecule(s) by reducing byproduct(s), focusing upon the specific example of stabilizing ethylene production in engineered microorganisms.

Reply: The authors appreciate the referee for her/his time on reviewing our work and appreciate the positive comments.

REVIEWERS' COMMENTS

Reviewer #1 (Remarks to the Author):

Thank you for addressing my concerns. Although this guanidinase appears to be a slow enzyme, your results are convincing. I also appreciate that you have established the whole stoichiometry of the reaction by determining the amount of ammonium produced together with urea. It is great that an activity of the enzyme encoded in this ORF (SII1077) has been established.

Reviewer #2 (Remarks to the Author):

I find no further issues with the revised version of the paper, and think it can be published. I thank the authors for their thoughtful responses to my earlier comments.